# GRADIENT RECTIFICATION FOR ROBUST CALIBRATION UNDER DISTRIBUTION SHIFT

## ABSTRACT

Deep neural networks often produce overconfident predictions, undermining their reliability in safety-critical visual applications. This miscalibration is further exacerbated under test distribution shift. Existing methods improve calibration via training-time regularization or post-hoc adjustment, but often rely on access to (or simulation of) target domains, limiting practicality. We propose Frequency-aware Gradient Rectification (FGR), a target-agnostic training framework for robust calibration. From a frequency perspective, FGR applies low-pass filtering to a subset of training images to diminish spurious high-frequency cues and bias learning toward domain-invariant structure. However, the associated information loss can degrade In-Distribution (ID) calibration. To resolve this trade-off, FGR treats ID calibration as a hard optimization constraint and rectifies parameter updates via geometric projection whenever they conflict with calibration. This projection-based update guarantees a first-order non-increase of the ID calibration objective without introducing additional weighting hyperparameters. Experiments on CIFAR-10/100-C and WILDS show that FGR significantly improves calibration under diverse shifts while preserving ID performance, and it remains compatible with post-hoc temperature scaling.

## 1 INTRODUCTION

Deep learning models have achieved remarkable accuracy across numerous tasks. However, in high-stakes applications such as autonomous driving (Cao et al., 2024) and clinical diagnostics (Penso et al., 2024), it is equally critical that these models provide reliable confidence estimates alongside accurate predictions, as overconfident errors can lead to catastrophic consequences.

Calibration performance quantifies the alignment between a model's predicted confidence and its true accuracy, e.g., among predictions with 0.6 confidence, approximately 60% should be correct. Unfortunately, modern deep learning models are commonly overconfident on incorrect predictions (Guo et al., 2017), leading to poor calibration. This problem becomes more severe in real-world deployments, where models inevitably encounter distribution shifts—test inputs that differ from the training distribution due to variations in lighting, weather, image quality, and other environmental factors (Ovadia et al., 2019). Therefore, it is essential to ensure that models remain well-calibrated not only in-distribution but also under a wide range of distribution shifts, as illustrated in Figure 1.

Existing approaches addressing this issue can be broadly divided into Target-Domain-Aware and Target-Domain-Agnostic methods. Target-Domain-Aware methods leverage information from target domains or domain generation rules to learn adaptive calibration strategies (Tomani et al., 2021; Yu et al., 2022). These approaches either utilize known multi-domain data to train input-specific calibration functions, or construct validation sets that simulate target domain characteristics for temperature scaling (Guo et al., 2017). However, because these methods depend on explicit or simulated target domain information, their practical applicability in real-world scenarios is limited, particularly when faced with diverse and unknown distribution shifts.

In contrast, Target-Domain-Agnostic methods do not require access to target domain information. Instead, they implicitly suppress overconfidence under distribution shifts though modifications made at training time, including two main paradigms: (1) Calibration-aware loss functions. For example, Focal Loss reweights easy examples (Mukhoti et al., 2020), and MaxEnt Loss encourages predictions to stay close to the statistical patterns observed during training, reducing overreaction to unseen

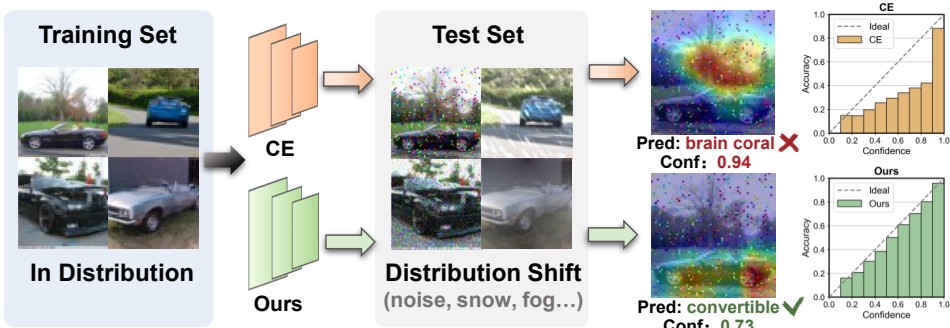

Figure 1: When deployed models encounter test data from a shifted distribution, traditional methods are often affected by distribution changes, leading to overly confident mispredictions, while our method focuses on domain-invariant features to provide well-calibrated outputs.

shifts (Neo et al., 2024). (2) Regularization techniques such as label smoothing (Müller et al., 2019) and Mixup (Thulasidasan et al., 2019), which discourage the model from producing overly sharp predictive distributions. While these methods can mitigate miscalibration, they lack explicit mechanisms to handle distribution shifts, often providing only indirect benefits in such scenarios.

We tackle these limitations by introducing a novel training framework that enhances calibration under distribution shift without relying on target domain information. Since distribution shifts often alter high-frequency visual patterns, which deep models tend to exploit as shortcut cues (Karimi & Gholipour, 2022), models may exploit these shortcuts to reduce predictive entropy, leading to overconfidence—for instance, learning to recognize "birds" based on special texture (e.g., green leafy patterns) rather than shape. Motivated by this, we apply Discrete Cosine Transform (DCT) filtering to isolate low-frequency image components, encouraging the model to rely on shape-related information that is more consistent across distributions. However, due to information loss, training on filtered images may also disrupt the fine-grained decision boundaries needed for ID performance, resulting in insufficient confidence in decisions.

To resolve this critical trade-off, we propose a training-time gradient rectification strategy that treats ID calibration as a hard constraint during optimization. Specifically, we train the model on a hybrid input set combining original and filtered images. In each step, we compute the geometric relationship between the main loss gradient (e.g., Focal Loss) on the hybrid batch and the ID calibration loss gradient (e.g., Soft-ECE (Karandikar et al., 2021)). When these gradients conflict (point in opposite directions), we project the main gradient onto the hyperplane orthogonal to the calibration gradient, ensuring under a first-order approximation that the update does not increase the ID calibration loss. This parameter-free intervention significantly improves calibration under distribution shifts while preserving ID performance. Overall, our contributions are summarized as follows:

- We explore calibration under distribution shift from a frequency-domain perspective and introduce a low-frequency filtering strategy to encourage reliance on domain-invariant features, improving shift calibration without access to target domain information.

- We propose a gradient rectification mechanism that treats in-distribution calibration as a hard first-order constraint and enforces it through geometric alignment during training, effectively balancing calibration across domains.

- Extensive experiments on synthetic and real-shift datasets showed that our method significantly improves calibration under shift while preserving strong ID performance.

## 2 RELATED WORK

**Uncertainty Calibration.** Approaches to improve the calibration are typically categorized into post-hoc and training-time methods. Post-hoc methods are applied to pre-trained models without altering their weights. A widely used baseline is Temperature Scaling (TS) (Guo et al., 2017), which optimizes a single scalar to rescale logits. More flexible alternatives include $\rho$-Norm scaling (Zhang & Xie, 2025), which generalizes TS via norm-based adjustments, and isotonic regression (Zadrozny & Elkan, 2001), a non-parametric approach that fits a monotonic mapping between predicted con-

fidence and empirical accuracy. Tao et al. (2025) clips the feature magnitudes of overconfident samples to increase their predictive entropy. Training-time methods aim to learn calibrated models directly by modifying the loss function or optimization process. Representative approaches include MMCE (Kumar et al., 2018), AvUC (Krishnan & Tickoo, 2020), Soft-ECE (Karandikar et al., 2021) and Dual Focal Loss (Tao et al., 2023b), which directly penalize miscalibration or down-weight confident prediction. Lin et al. (2025) modulate the updated scaling gradient using the uncertainty of each sample. In addition, regularization techniques such as Label Smoothing (Müller et al., 2019), Mixup (Zhang et al., 2021) and CutMix (Yun et al., 2019) can provide indirect calibration benefits.

**Calibration under Distribution Shift.** While the aforementioned methods perform well in-distribution, their calibration performance is often fragile to distribution shifts (Ovadia et al., 2019). To address this, adaptive temperature scaling methods (Yu et al., 2022; Wang et al., 2024; Choi et al., 2024) train regressors using augmented validation sets or known auxiliary domains to estimate input-specific temperature. Other methods incorporate feature density (Tomani et al., 2023), Bayesian inference (Seligmann et al., 2023), or prior training states (Tao et al., 2023a) to enhance calibration under shift, often at the cost of additional computation or assumptions. Data augmentation can also improve robustness by exposing the model to varied inputs (Hendrycks et al., 2020). However, these methods often rely on auxiliary domains, model priors, or handcrafted regularizers—limiting their applicability under unknown or dynamic shifts.

**Frequency-Domain Robustness.** Recent frequency-domain studies reveal insights into model robustness. Yin et al. (2019) show that models often exploit high-frequency non-robust statistics, while Fridovich-Keil et al. (2022) reveal model sensitivity to spectral characteristics of input images. Li et al. (2023) demonstrate that discarding high-frequency components can preserve semantically meaningful information through DCT. Building upon these insights, our work leverages frequency filtering to build inherent distribution shift calibration robustness without any target information.

## 3 PROBLEM FORMULATION

We consider a image classification task over a dataset $\mathcal{D} = \{(\boldsymbol{x}_i, y_i)\}_{i=1}^N$ with $N$ samples, where $\boldsymbol{x}_i \in \mathcal{X}$ represents the input and $y_i \in \{1, 2, \ldots, K\}$ denotes the ground-truth class label for $K$ classes. A neural network $f(\theta)$ with parameters $\theta$ maps input $\boldsymbol{x}_i$ to logits $\boldsymbol{z}_i = f(\boldsymbol{x}_i; \theta)$. After applying the Softmax function, the predicted probability for class $k$ is given by $p_{ik} = \frac{\exp(z_{ik})}{\sum_{j=1}^K \exp(z_{ij})}$. The predicted class label $\hat{y}_i$ and corresponding confidence $\hat{p}_i$ are defined as:

$$\hat{y}_i = \arg\max_k p_{ik}, \ \ \hat{p}_i = \max_k p_{ik}, \ k \in \{1, \ldots, K\}. \tag{1}$$

A model is perfectly calibrated if its confidence scores accurately reflect the true likelihood of correctness, formally satisfying $P(\hat{y} = y | \hat{p} = p) = p$ for all $p \in [0, 1]$. In practice, since the true posterior distribution is unknown, this ideal condition is approximated by partitioning predictions into bins based on confidence levels (Gawlikowski et al., 2023).

**Expected Calibration Error (ECE):** ECE is the most widely-used metric to quantify calibration performance. The confidence interval $[0, 1]$ is partitioned into $M$ bins $\{B_m\}_{m=1}^M$, where bin $B_m$ contains all samples with confidence $\hat{p} \in \left(\frac{m-1}{M}, \frac{m}{M}\right]$. Then, ECE is calculated as the weighted average of absolute differences between accuracy and confidence across all bins:

$$\text{ECE} = \sum_{m=1}^M \frac{|B_m|}{N} |\text{acc}(B_m) - \text{conf}(B_m)|. \tag{2}$$

where $\text{acc}(B_m)$ and $\text{conf}(B_m)$ represent the empirical accuracy and average confidence within bin $B_m$, respectively. Additional calibration metrics such as Classwise ECE (CECE) are discussed in the Appendix C.1.

**Distribution Shift Calibration:** In real-world deployment, models encounter distribution shifts where test data $\mathcal{D}_{\text{test}}$ differs from training data $\mathcal{D}_{\text{train}}$. Our goal is to learn a model that maintains well-calibrated predictions on both in-distribution data and under various distribution shifts, without requiring access to target domain information during training.

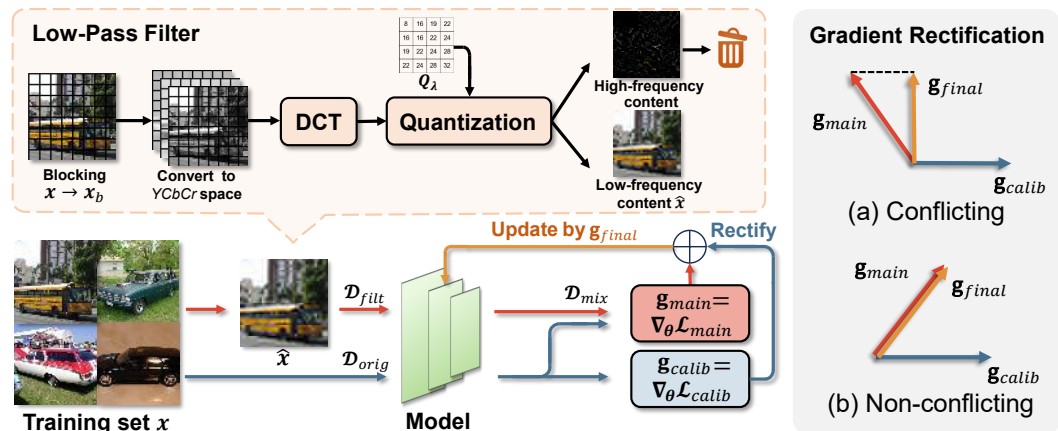

Figure 2: Overview of the proposed method. We apply DCT-based low-pass filtering to a subset of training data to suppress high-frequency features and construct a hybrid dataset $\mathcal{D}_{\text{mix}}$. During optimization, we compute gradient $\mathbf{g}_{\text{main}}$ on $\mathcal{D}_{\text{mix}}$ and $\mathbf{g}_{\text{calib}}$ on $\mathcal{D}_{\text{orig}}$, and apply gradient rectification when they conflict to prevent degradation of ID calibration.

## 4 METHOD

From a frequency perspective, we first use low-pass filtering to enhance robustness under distribution shift. However, this introduces a trade-off, as filtering can degrade ID calibration by causing underconfidence. We resolve this with a gradient rectification mechanism that treats ID calibration as a hard, first-order constraint, ensuring robustness gains do not compromise ID performance.

### 4.1 LOW-PASS FILTERING FOR ROBUST FEATURES

Prior work has shown that distribution shifts often alter high-frequency visual patterns that models exploit as predictive shortcuts (Fridovich-Keil et al., 2022; Li et al., 2023). This can lead to overconfident predictions based on features with spurious correlations. Motivated by this, we apply a low-pass filter to suppress source-specific high-frequency cues, encouraging reliance on domain-invariant semantic features under shift. We specifically choose a Discrete Cosine Transform (DCT)-based approach (Khayam, 2003) for its strong energy compaction property and block-wise processing, which avoids the global ringing artifacts of Fourier-based methods. This makes it highly effective for preserving core semantics while robustly discarding high-frequency noise.

At the beginning of each training epoch, we randomly select a proportion $\rho$ of the training samples and apply the low-pass filtering to obtain a filtered subset $\mathcal{D}_{\text{filt}}$. The remaining $(1-\rho)$ portion of samples are kept unchanged, forming the unfiltered subset $\mathcal{D}_{\text{orig}}$. We then define the hybrid training set as $\mathcal{D}_{\text{mix}} = \mathcal{D}_{\text{filt}} \cup \mathcal{D}_{\text{orig}}$.

For the filtering process, we implement the DCT approach using a block-wise method. Given an input image $\boldsymbol{x} \in \mathbb{R}^{H \times W \times 3}$, we first convert it to the YCbCr color space. Each channel is then partitioned into non-overlapping $8 \times 8$ blocks. This local processing is robust to common texture distortions without introducing global artifacts. We apply a 2D-DCT to each block $\boldsymbol{x}_b$:

$$\mathbf{F}_b = \text{DCT}(\boldsymbol{x}_b), \tag{3}$$

where $\mathbf{F}_b \in \mathbb{R}^{8 \times 8}$ contains the frequency coefficients. These coefficients are then quantized by dividing element-wise with the given quantization matrix $\mathbf{Q}_\lambda$ and rounding to the nearest integer:

$$\mathbf{F}_b^{(\text{q})} = \text{round}\left(\frac{\mathbf{F}_b}{\mathbf{Q}_\lambda}\right), \tag{4}$$

where $\lambda \in [1, 100]$ controls the filtering strength. $\mathbf{Q}_\lambda$ is obtained by scaling standard JPEG tables, making the filtering intensity easily adjustable. A lower $\lambda$ corresponds to more aggressive filtering. The quantized coefficients are then de-quantized and inversely transformed to reconstruct the filtered block:

$$\hat{\mathbf{F}}_b = \mathbf{F}_b^{(\text{q})} \cdot \mathbf{Q}_\lambda, \quad \hat{\boldsymbol{x}}_b = \text{DCT}^{-1}\left(\hat{\mathbf{F}}_b\right). \tag{5}$$

The final filtered image $x'$ is formed by reassembling all $\hat{x}_b$ blocks and converting back to the RGB space. This hybrid strategy exposes the model to both original full-spectrum inputs and low-pass filtered versions, discouraging reliance on domain-specific artifacts.

## 4.2 GRADIENT RECTIFICATION

While low-pass filtering encourages the model to learn domain-invariant features and improves robustness under distribution shift, it may simultaneously degrade ID calibration. As shown in the reliability diagrams in Figure 3, more aggressive filtering (i.e., a lower $\lambda$) removes fine-grained cues, causing the model to become underconfident on ID data. To address this trade-off, we propose a gradient rectification mechanism that treats ID calibration as a hard constraint during optimization, rather than a competing objective. This ensures that updates aimed at improving shift robustness do not compromise ID calibration. Specifically, our approach manages two potentially conflicting objectives:

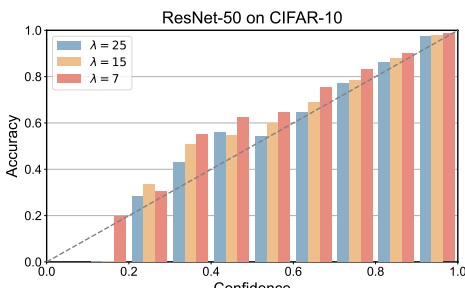

Figure 3: Effect of filtering strength $\lambda$ on ID calibration. Lower $\lambda$ (more aggressive filtering) leads to underconfidence.

- **Generalization Objective:** We aim to learn robust features through a main classification loss $\mathcal{L}_{\text{main}}$ (e.g., Cross Entropy or Focal Loss) computed on the mixed dataset $\mathcal{D}_{\text{mix}}$ containing both original and frequency-filtered images. The gradient for this objective is:

$$\mathbf{g}_{\text{main}} = \nabla_\theta \mathcal{L}_{\text{main}}(\theta; \mathcal{D}_{\text{mix}}). \tag{6}$$

- **ID Calibration Objective:** We preserve ID calibration through a explicit calibration loss $\mathcal{L}_{\text{calib}}$ (e.g., Soft-ECE (Karandikar et al., 2021)) computed exclusively on original training data $\mathcal{D}_{\text{orig}}$. The objective's gradient is:

$$\mathbf{g}_{\text{calib}} = \nabla_\theta \mathcal{L}_{\text{calib}}(\theta; \mathcal{D}_{\text{orig}}). \tag{7}$$

When these two gradients conflict (i.e., point in opposite directions), we rectify the main gradient by projecting it onto the hyperplane orthogonal to the calibration gradient. Otherwise, if the gradients are aligned, the update proceeds normally, as illustrated in Figure 2 (right). Formally, final rectified gradient $\mathbf{g}_{\text{final}}$ define as:

$$\mathbf{g}_{\text{final}} = \begin{cases} \mathbf{g}_{\text{main}}, & \text{if } \mathbf{g}_{\text{main}} \cdot \mathbf{g}_{\text{calib}} \geq 0 \\ \mathbf{g}_{\text{main}} - \frac{\mathbf{g}_{\text{main}} \cdot \mathbf{g}_{\text{calib}}}{\|\mathbf{g}_{\text{calib}}\|^2 + \epsilon} \mathbf{g}_{\text{calib}}, & \text{otherwise,} \end{cases} \tag{8}$$

where $\epsilon$ is a small constant for numerical stability. This projection guarantees a non-increase of the ID calibration loss $\mathcal{L}_{\text{calib}}$ to a first-order approximation, effectively navigating the Pareto front of the two objectives without introducing trade-off hyperparameters (Yu et al., 2020; Zhu et al., 2023).

In our implementation, we adopt Dual Focal Loss (DFL) (Tao et al., 2023b) as the main objective $\mathcal{L}_{\text{main}}$ over the mixed dataset $\mathcal{D}_{\text{mix}}$. DFL introduces a dual modulation mechanism that penalizes both underconfident and overconfident predictions, yielding better calibration potential than standard cross-entropy. For a sample $x$, its formulation is:

$$\mathcal{L}_{\text{main}} = -\sum_{k=1}^{K} y_k \left(1 - \hat{p}_k(x) + \hat{p}_j(x)\right)^\gamma \log \hat{p}_k(x), \tag{9}$$

where $j$ denotes the highest-scoring incorrect class and $\gamma$ is a tunable focusing parameter.

To supervise ID calibration, we employ Soft-Binned ECE (Soft-ECE) (Karandikar et al., 2021) as the calibration loss $\mathcal{L}_{\text{calib}}$, computed on unfiltered samples $\mathcal{D}_{\text{orig}}$. Soft-ECE provides a differentiable approximation of the standard ECE by using a soft, temperature-controlled binning function. Its general form is:

$$\mathcal{L}_{\text{calib}} = \left( \sum_{m=1}^{M} \frac{|S_m|}{N} |\text{acc}(S_m) - \text{conf}(S_m)|^2 \right)^{1/2}, \tag{10}$$

where $S_m$ are soft bins derived from a membership function. A detailed description of the Soft-ECE formulation and its implementation is provided in Appendix C.4.2.

**Generalization Error Analysis.** We provide theoretical justification for our gradient rectification mechanism by analyzing its generalization error bound. Our approach modifies standard empirical risk minimization by enforcing a constraint that updates driven by the main loss over the hybrid dataset $\mathcal{D}_{\text{mix}}$ must not degrade calibration performance on the original dataset $\mathcal{D}_{\text{orig}}$. This constrained optimization implicitly regularizes model updates through projection-based correction, leading to tighter generalization guarantees compared to naive training on $\mathcal{D}_{\text{mix}}$ alone.

Let $\mathcal{R}(\cdot)$ and $\hat{\mathcal{R}}(\cdot)$ denote the expected and empirical risks, respectively. A naive model $\hat{f}_{\text{naive}}$ trained only on the mixed dataset minimizes $\hat{f}_{\text{naive}} = \arg\min_{f \in \mathcal{F}} \hat{\mathcal{R}}_{\text{mix}}(f)$, where $\mathcal{F}$ is the function class. Our gradient rectification strategy can be viewed as optimizing a joint empirical risk:

$$\hat{f}_{\text{FGR}} = \arg\min_{f \in \mathcal{F}} \left( \hat{\mathcal{R}}_{\text{mix}}(f) + \hat{\mathcal{R}}_{\text{calib}}(f) \right) \tag{11}$$

where $\hat{\mathcal{R}}_{\text{calib}}$ represents the empirical calibration risk on $\mathcal{D}_{\text{orig}}$. We bound the generalization error using *Rademacher Complexity* theory (Bartlett & Mendelson, 2002) and the Theorem 6.2 in Zhang et al. (2012).

**Theorem 1.** Let $\mathcal{D}_{\text{mix}}$ and $\mathcal{D}_{\text{orig}}$ be the distributions for the mixed and original datasets, with $N_{\text{mix}}$ and $N_{\text{orig}}$ i.i.d. samples drawn from them, respectively. Then for any $\delta > 0$, with probability at least $1 - \delta$, the expected risk on the primary objective for our model $\hat{f}_{\text{FGR}}$ is bounded by:

$$\mathcal{R}_{\text{mix}}(\hat{f}_{\text{FGR}}) \leq \hat{\mathcal{R}}_{\text{mix+calib}}(\hat{f}_{\text{FGR}}) + \frac{1}{2}\mathcal{W}_{\mathcal{F}}(\mathcal{D}_{\text{mix}}, \mathcal{D}_{\text{orig}}) + \mathfrak{C}(\mathcal{F}, N_{\text{mix}}, N_{\text{orig}}, \delta), \tag{12}$$

where $\mathcal{W}_{\mathcal{F}}(\mathcal{D}_{\text{mix}}, \mathcal{D}_{\text{orig}})$ captures the distribution discrepancy introduced by our controlled frequency filtering, and $\mathfrak{C}$ is a complexity term depending on the Rademacher complexities of $\mathcal{F}$ under the two data sources, and vanishes as $N_{\text{mix}}, N_{\text{orig}} \to \infty$.

In contrast, the generalization error for the naive model is bounded by:

$$\mathcal{R}_{\text{mix}}(\hat{f}_{\text{naive}}) \leq \hat{\mathcal{R}}_{\text{mix}}(\hat{f}_{\text{naive}}) + \mathfrak{C}'(\mathcal{F}, N_{\text{mix}}, \delta), \tag{13}$$

where $\mathfrak{C}'$ depends only on $N_{\text{mix}}$. When empirical risks are effectively minimized, comparing the dominant terms in our bounds reveals the advantage. Our method achieves a generalization gap of approximately $\mathcal{W}_{\mathcal{F}}/2 + \mathfrak{C}(\mathcal{F}, N_{\text{mix}}, N_{\text{orig}}, \delta)$, while the naive baseline suffers from $\mathfrak{C}'(\mathcal{F}, N_{\text{mix}}, \delta)$. Since our constraint mechanism provides additional regularization through the calibration objective, and $\mathcal{W}_{\mathcal{F}}$ is small due to controlled filtering, our bound is provably tighter, providing theoretical justification for the empirical superiority of our approach. The detailed proof is in Appendix E.

## 5 EXPERIMENTS

We evaluate the effectiveness of our proposed method on both synthetic and real-world distribution shift benchmarks, with a focus on improving calibration without sacrificing classification accuracy. Our experiments are designed to demonstrate: (1) our method's improvement in calibration under distribution shift; (2) its ability to maintain strong calibration performance on clean in-distribution data; (3) its influence on guiding the model to focus on semantically meaningful features; and (4) the individual contributions of our method's key components through an ablation study.

### 5.1 EXPERIMENTAL SETUP

**Datasets.** We evaluate our method on a diverse set of benchmarks. For **synthetic shifts**, we use CIFAR-10/100 and Tiny-ImageNet as in-distribution (ID) data, with their corresponding corrupted versions (CIFAR-10/100-C, Tiny-ImageNet-C) (Hendrycks & Dietterich, 2019) serving as distribution shift test sets. These corrupted datasets cover 15 common corruption types across 5 severity levels. For **real-world shifts**, we use Camelyon17, iWildCam, and FMoW from the WILDS benchmark (Koh et al., 2021), which feature naturally occurring domain shifts from different hospitals, camera traps, and geographic regions, respectively.

**Compared Methods.** We compare our training-time method against a suite of strong baselines: standard Cross-Entropy (CE), Label Smoothing (LS-0.05) (Müller et al., 2019), Mixup (Zhang et al., 2021), AugMix (Hendrycks et al., 2020), FLSD-53 (Mukhoti et al., 2020), Dual Focal Loss (DFL) (Tao et al., 2023b), MaxEnt (Neo et al., 2024), and BSCE-GRA (Lin et al., 2025). We also report results with and without post-hoc Temperature Scaling (TS) (Guo et al., 2017) to evaluate compatibility.

**Implementation Details.** For experiments on CIFAR and Tiny-ImageNet, we follow the setup in Mukhoti et al. (2020), training ResNet-50/110, DenseNet-121, and Wide-ResNet-26 models from scratch for 350 epochs. For our FGR method, we also train from scratch, but introduce the frequency filtering and gradient rectification starting from epoch 200. This allows the model to first establish robust classification boundaries before applying our calibration-focused optimizations. For WILDS datasets, all methods fine-tune ImageNet pre-trained models following the official WILDS training protocols. To demonstrate the practical benefits of our approach, we also conducted experiments using a two-stage fine-tuning strategy, which yields comparable or even superior performance with significantly reduced computational cost. Detailed hyperparameters for all methods and datasets, along with specifics of the fine-tuning experiments, are provided in Appendix C.

## 5.2 PERFORMANCE UNDER DISTRIBUTION SHIFT

We evaluate our method on both synthetic corruption datasets and real-world distribution shift benchmarks. Table 1 summarizes the results across multiple datasets and architectures.

On synthetic shifts like CIFAR-10-C and CIFAR-100-C, our approach significantly reduces ECE compared to strong baselines like MaxEnt M and BSCE-GRA. This demonstrates that encouraging reliance on domain-invariant features improves robustness. Unlike methods with aggressive augmentation such as AugMix, which perform well on synthetic data but poorly on real-world WILDS datasets, our method also generalizes effectively to real-world shifts. It outperforms all baselines on Camelyon17 in both ECE (2.36%) and CECE (5.71%), and remains competitive on FMoW and iWildCam. Furthermore, Figure 4 shows that our method maintains low ECE across all corruption severities, especially in high-shift scenarios. The results in Table 1 ("w/ TS" columns) also confirm that our method is compatible with post-hoc temperature scaling, often leading to further calibration improvements.

Table 1: Test scores (%) of different methods on synthetic (top) and real-world (bottom) distribution shift test sets. For synthetic datasets, results are averaged over 15 corruption types across 5 severity levels. The "w/ TS" columns show ECE and CECE values with temperature scaling post-hoc calibration. The best average scores are highlighted in **bold**.

| Loss Fn. | CIFAR10-C / DenseNet-121 | | | | | CIFAR100-C / DenseNet-121 | | | | | Tiny ImageNet-C / DenseNet-121 | | | | |
|---|---|---|---|---|---|---|---|---|---|---|---|---|---|---|---|
| | ACC. | ECE | w/ TS | CECE | w/ TS | ACC. | ECE | w/ TS | CECE | w/ TS | ACC. | ECE | w/ TS | CECE | w/ TS |
| CE | 74.05 | 23.51 | 17.16 | 4.87 | 3.86 | 48.44 | 41.21 | 13.31 | 0.91 | 0.47 | 24.27 | 25.83 | 36.23 | 0.44 | 0.52 |
| LS-0.05 | 73.65 | 16.42 | 18.36 | 3.81 | 4.07 | 51.37 | 19.47 | 16.20 | 0.55 | 0.52 | 25.96 | 16.11 | 16.11 | **0.37** | **0.37** |
| Mixup | 75.41 | 14.80 | 17.81 | 3.79 | 3.97 | 53.38 | 12.43 | 16.32 | 0.50 | 0.53 | 26.26 | **11.75** | 16.77 | **0.37** | 0.40 |
| AugMix | 82.49 | 11.03 | **8.51** | 3.31 | **2.83** | 59.87 | 16.14 | 7.84 | 0.50 | **0.45** | 14.51 | 22.55 | 11.98 | 0.48 | 0.40 |
| FLSD-53 | 72.61 | 13.58 | 14.99 | 3.74 | 3.86 | 49.39 | 13.74 | 10.04 | 0.56 | 0.53 | 22.30 | 15.35 | 47.58 | 0.41 | 0.63 |
| DFL | 70.18 | 16.19 | 15.12 | 4.28 | 4.19 | 50.17 | 9.97 | 8.82 | 0.51 | 0.50 | 23.84 | 12.23 | 16.62 | 0.38 | 0.40 |
| MaxEnt M | 71.98 | 11.62 | 13.63 | 3.62 | 3.79 | 48.34 | 11.05 | 10.38 | 0.57 | 0.57 | 21.14 | 21.79 | 17.05 | 0.46 | 0.44 |
| BSCE-GRA | 72.46 | 11.45 | 14.11 | 3.64 | 3.82 | 49.22 | 11.69 | 10.86 | 0.54 | 0.54 | 21.47 | 13.08 | 19.96 | 0.40 | 0.44 |
| Ours | 75.12 | **9.02** | 9.90 | **3.12** | 3.16 | 52.66 | **8.53** | 7.57 | **0.47** | 0.46 | 24.03 | 12.39 | **11.16** | 0.40 | 0.38 |

| Loss Fn. | Camelyon17 / DenseNet-121 | | | | | iWildCam / ResNet-50 | | | | | Fmow / DenseNet-121 | | | | |
|---|---|---|---|---|---|---|---|---|---|---|---|---|---|---|---|
| | ACC. | ECE | w/ TS | CECE | w/ TS | ACC. | ECE | w/ TS | CECE | w/ TS | ACC. | ECE | w/ TS | CECE | w/ TS |
| CE | 86.83 | 12.23 | 6.18 | 12.61 | 10.57 | 77.51 | 14.99 | 3.61 | 0.196 | 0.184 | 52.31 | 41.94 | 5.70 | 1.42 | 0.69 |
| LS-0.05 | 85.86 | 8.26 | 7.55 | 13.86 | 13.73 | 77.39 | 8.60 | 5.98 | 0.224 | 0.205 | 50.89 | 33.64 | 6.91 | 1.14 | 0.81 |
| Mixup | 88.33 | 2.14 | 2.05 | 10.09 | 10.00 | 75.85 | 2.40 | 3.88 | 0.164 | 0.172 | 51.53 | 23.16 | 5.53 | 0.86 | 0.65 |
| AugMix | 70.99 | 16.99 | 8.19 | 19.68 | 15.87 | 49.07 | 22.18 | 3.36 | 0.409 | 0.454 | 27.47 | 48.68 | 3.08 | 1.86 | 0.91 |
| FLSD-53 | 87.10 | 6.33 | 3.67 | 11.79 | 11.53 | 74.04 | 7.82 | 7.69 | 0.223 | 0.273 | 53.81 | 28.72 | 4.09 | 1.03 | 0.55 |
| DFL | 88.03 | 2.74 | 2.12 | 9.96 | 9.88 | 73.52 | 6.97 | 3.57 | 0.225 | 0.263 | 53.55 | 26.59 | 4.30 | 0.97 | 0.56 |
| MaxEnt M | 85.67 | 3.96 | 2.71 | 12.93 | 12.74 | 75.11 | 7.14 | 3.08 | 0.196 | 0.210 | 53.04 | 30.69 | 4.85 | 1.09 | 0.58 |
| BSCE-GRA | 86.44 | 5.53 | 2.50 | 11.34 | 10.98 | 75.19 | 5.02 | 3.74 | **0.150** | 0.181 | 53.53 | 27.40 | **3.60** | 0.99 | 0.55 |
| Ours | 89.19 | **2.36** | **1.82** | **5.71** | **5.38** | 76.11 | **3.34** | **2.97** | 0.160 | **0.179** | 51.95 | 25.06 | 3.84 | **0.92** | **0.53** |

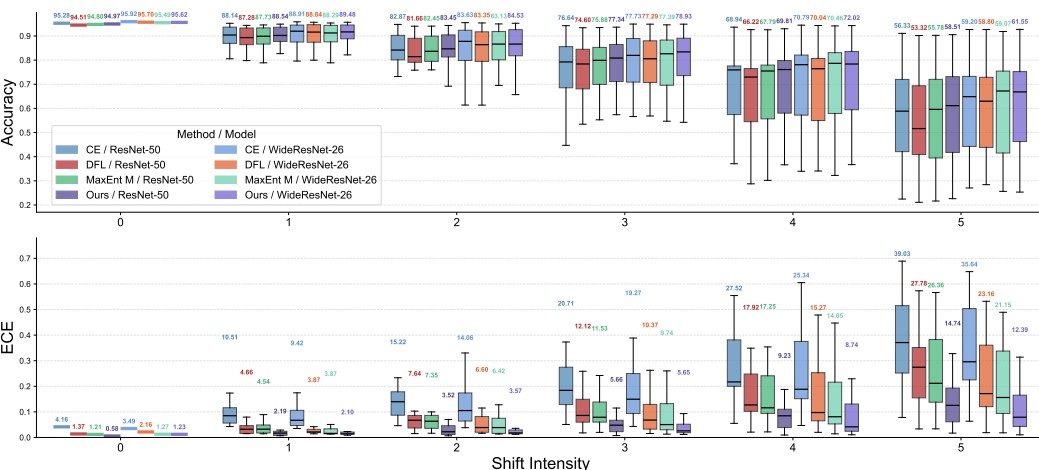

Figure 4: Test Accuracy↑ (%) and ECE↓ (%) of different methods trained on CIFAR-10 across ResNet-50 and WideResNet-26. Results are evaluated on clean test data and corrupted test sets with intensity levels 1-5. Each box shows quartiles summarizing results across all 15 corruption types. Numbers above boxes indicate the mean values across all corruption types.

Table 2: ECE↓ (%) before and after temperature scaling for different methods on the in-distribution test set. In the experiment, ECE was evaluated for different methods before (Pre T) and after (Post T) temperature scaling.

| Dataset | Model | CE | | FLSD-53 | | DFL | | MaxEnt | | BSCE-GRA | | Ours | |
|---|---|---|---|---|---|---|---|---|---|---|---|---|---|
| | | Pre T | Post T | Pre T | Post T | Pre T | Post T | Pre T | Post T | Pre T | Post T | Pre T | Post T |
| Cifar-10 | ResNet-50 | 4.16 | 1.14 | 1.28 | 1.01 | 1.37 | 1.28 | 1.42 | 1.49 | **0.60** | 0.70 | 0.65 | 0.68 |
| | ResNet-110 | 4.46 | 1.17 | 1.26 | 0.94 | 1.49 | 0.92 | 1.28 | 1.28 | 1.16 | 1.07 | 0.90 | **0.84** |
| | DenseNet-121 | 4.62 | 1.75 | 1.19 | 0.98 | 1.18 | 0.65 | 1.07 | 1.18 | 1.56 | 1.05 | 0.97 | **0.64** |
| | WideResnet-26 | 3.49 | 1.26 | 2.16 | 1.19 | 1.87 | 1.11 | 1.27 | 1.05 | 2.48 | 1.16 | 1.23 | **0.58** |
| Cifar-100 | ResNet-50 | 17.11 | 2.31 | 4.71 | 2.41 | 1.48 | 1.48 | 4.56 | 3.51 | 2.91 | **0.99** | 2.84 | 2.49 |
| | ResNet-110 | 18.28 | 4.27 | 6.75 | 3.99 | 3.27 | 3.27 | 3.72 | 3.94 | 3.85 | **2.96** | 3.48 | 3.57 |
| | DenseNet-121 | 18.95 | 3.57 | 2.93 | **1.42** | 1.96 | 1.51 | 1.71 | 1.75 | 1.68 | 1.54 | 3.32 | 3.34 |
| | WideResnet-26 | 14.75 | 2.86 | 2.06 | 2.41 | 2.77 | **1.64** | 2.38 | 2.48 | 2.00 | 1.86 | 3.01 | 2.93 |
| Tiny-ImageNet | ResNet-50 | 13.60 | 2.91 | 1.92 | 8.67 | 2.82 | **1.57** | 7.75 | 1.82 | 2.05 | 15.15 | 2.64 | 7.43 |
| | DenseNet-121 | 8.26 | 15.01 | 3.81 | 15.17 | 3.68 | 1.37 | 4.12 | 1.32 | 6.01 | 1.62 | 3.42 | **1.22** |

## 5.3 IN-DISTRIBUTION CALIBRATION PERFORMANCE

To ensure that our method does not harm calibration on clean in-distribution data, we evaluate all methods on the original test sets of CIFAR-10/100 and Tiny-ImageNet. As shown in Table 2, our method maintains competitive or superior ECE performance in most cases, even in the absence of distribution shifts, demonstrating its general applicability.

## 5.4 VISUALIZATION OF MODEL FOCUS

To understand how our method improves calibration, we visualize model attention patterns and reliability on iWildCam using Grad-CAM (Selvaraju et al., 2017). Figure 5 shows the comparison between BSCE-GRA and our method. The left panel reveals significant differences in model focus. For the top image, while both methods make correct predictions, BSCE-GRA primarily attends to background noise rather than the monkey itself, whereas our method focuses directly on the animal. In the bottom case, BSCE-GRA's attention on irrelevant regions leads to an incorrect prediction, while our method correctly identifies the target.

The reliability diagram shows that the predictions generated by our method are closer to the ideal diagonal line, indicating better calibration performance. These visualizations confirm that our frequency filtering successfully guides the model to rely on semantically meaningful features rather

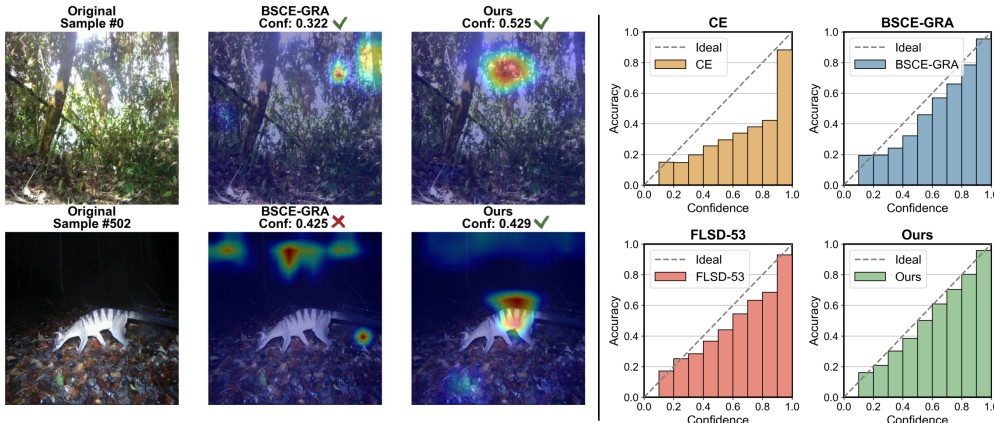

Figure 5: Visualization comparison on iWildCam dataset. Left: Grad-CAM attention maps for BSCE-GRA and our method. Right: Reliability diagrams for both methods.

than spurious high-frequency patterns, resulting in both better accuracy and more reliable confidence estimates.

## 5.5 ABLATION STUDY

We perform an ablation study to examine the roles of frequency filtering and gradient rectification. Table 3 compares the FGR method with three variants: (1) **Filter Only**, which applies filtering without rectification; (2) **Rect. Only**, which applies rectification without filtering; and (3) **Weighted Sum**, which combines the main and calibration losses using a fixed weight instead of our projection-based rectification.

The results highlight a critical trade-off. Filter Only improves calibration under some distribution shifts (e.g., ECE on CIFAR-100-C is 9.70) but severely degrades in-distribution (ID) performance (ECE on CIFAR-10 is 11.54). Conversely, Rect. Only improves ID calibration but fails under shift. The Weighted Sum approach can find a balance, achieving good results after careful tuning of the loss weight. However, our parameter-free FGR method consistently achieves a superior balance, delivering the best or near-best calibration on both ID (e.g., 0.65 ECE on CIFAR-10) and shift datasets (e.g., 6.78 ECE on CIFAR-10-C) without manual weight tuning. This demonstrates that both components are necessary and that our gradient rectification is more effective and convenient than simple weighted averaging for resolving the objectives' conflict.

Table 3: Ablation study results on In-Distribution (ID) and distribution shift test sets.

| Method | In-Distribution | | | | | | Distribution Shift | | | | | |
|---|---|---|---|---|---|---|---|---|---|---|---|---|
| | CIFAR-10 | | CIFAR-100 | | Tiny-INet | | CIFAR-10 | | CIFAR-100 | | Tiny-INet | |
| | ACC | ECE | ACC | ECE | ACC | ECE | ACC | ECE | ACC | ECE | ACC | ECE |
| Filter Only | 95.21 | 11.54 | 77.90 | 11.65 | 64.33 | 2.92 | 75.17 | 9.78 | 51.12 | **9.70** | 22.98 | 18.41 |
| Rect. Only | 95.19 | 2.24 | 78.21 | 2.96 | 64.02 | 7.76 | 74.40 | 18.60 | 51.07 | 27.52 | 22.71 | 26.62 |
| Weighted Sum | 95.03 | 1.72 | 77.82 | **2.35** | 64.28 | 3.02 | 75.37 | 7.93 | 51.15 | 11.80 | 22.64 | 17.26 |
| FGR | 94.97 | **0.65** | 78.30 | 2.84 | 64.18 | **2.64** | 75.23 | **6.78** | 50.89 | 10.50 | 23.25 | **15.46** |

## 6 CONCLUSION

We propose Frequency-aware Gradient Rectification (FGR), a training framework designed to address calibration degradation under distribution shift without requiring access to target domain data. By applying frequency-domain filtering, the model is encouraged to focus on domain-invariant features rather than spurious high-frequency patterns. Meanwhile, to preserve in-distribution calibration, we introduce a gradient rectification mechanism that treats in-distribution calibration as a hard constraint via geometric projection. Extensive experiments on synthetic and real-world benchmarks demonstrate that our method consistently delivers well-calibrated predictions without compromising classification accuracy, and remains compatible with post-hoc calibration methods.

ETHICS STATEMENT

This work adheres to the ICLR Code of Ethics. In this study, no human subjects or animal experimentation was involved. All datasets used, including CIFAR-10/100, Tiny-ImageNet, and the WILDS benchmark (Camelyon17, iWildCam, FMoW), are public benchmarks and were sourced in compliance with their respective usage guidelines, ensuring no violation of privacy. We have taken care to avoid any biases or discriminatory outcomes in our research process. No personally identifiable information was used, and no experiments were conducted that could raise privacy or security concerns. We are committed to maintaining transparency and integrity throughout the research process.

REPRODUCIBILITY STATEMENT

To ensure the reproducibility of our work, we have provided comprehensive details in the appendix. Section B of the appendix presents the pseudocode for our FGR algorithm. Section C details the experimental setup, including dataset preprocessing (Section C.2), model architectures and training configurations (Section C.3), and specific hyperparameter settings for both our method and all baseline methods (Section C.4). The complete theoretical proof for Theorem 1 is provided in Section E. We have submitted the source code in the supplementary materials and will make it publicly available upon publication.

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

## A    APPENDIX

This appendix provides comprehensive supplementary materials for our paper. We organize the content as follows:

- **Section B** presents the complete pseudocode for our Frequency-aware Gradient Rectification (FGR) method.

- **Section C** details the full experimental setup. This includes evaluation metrics (Section C.1), dataset descriptions and preprocessing steps (Section C.2), model architectures and training configurations (Section C.3), and implementation details for both our method and all baselines (Section C.4).

- **Section D** reports additional experimental results, including hyperparameter sensitivity analysis, a comparison of training strategies, and a computational complexity analysis.

- **Section E** provides the complete theoretical proof for Theorem 1.

## B    OVERALL ALGORITHM DESCRIPTION

Algorithm 1 summarizes our complete training procedure. The process begins with a warm-up phase, where the model is trained using only the main loss. This is followed by the FGR phase, where in each training step, we sample batches from both the mixed and original datasets to compute the main and calibration gradients, respectively, and apply gradient rectification when their directions conflict.

---

**Algorithm 1** Frequency-aware Gradient Rectification (FGR)

---

**Input:** Training set $\mathcal{D}$, filtering ratio $\rho$, main loss focusing parameter $\gamma$, FGR start epoch $E_{\text{start}}$
**Output**: Trained model parameters $\theta$

1: Initialize model parameters $\theta$
2: **for** epoch = 1 to Total Epochs **do**
3:     **if** epoch $< E_{\text{start}}$ **then**
4:         // Standard training phase (warm-up)
5:         **for** each batch $b$ from $\mathcal{D}$ **do**
6:             Compute main loss gradient $\mathbf{g}_{\text{main}} = \nabla_\theta \mathcal{L}_{\text{main}}(\theta; b)$
7:             Update $\theta$ using $\mathbf{g}_{\text{main}}$
8:         **end for**
9:     **else**
10:        // FGR training phase
11:        Randomly sample $\rho$ portion of $\mathcal{D}$ to create $\mathcal{D}_{\text{filt}}$ via DCT filtering
12:        Let $\mathcal{D}_{\text{orig}} = \mathcal{D} \setminus \mathcal{D}_{\text{filt}}$, and $\mathcal{D}_{\text{mix}} = \mathcal{D}_{\text{filt}} \cup \mathcal{D}_{\text{orig}}$
13:        **for** each training step **do**
14:            Sample a batch $b_{\text{mix}}$ from $\mathcal{D}_{\text{mix}}$
15:            Sample a batch $b_{\text{orig}}$ from $\mathcal{D}_{\text{orig}}$
16:            Compute main loss gradient $\mathbf{g}_{\text{main}} = \nabla_\theta \mathcal{L}_{\text{main}}(\theta; b_{\text{mix}})$
17:            Compute calibration loss gradient $\mathbf{g}_{\text{calib}} = \nabla_\theta \mathcal{L}_{\text{calib}}(\theta; b_{\text{orig}})$
18:            **if** $\mathbf{g}_{\text{main}} \cdot \mathbf{g}_{\text{calib}} < 0$ **then**
19:                $\mathbf{g}_{\text{final}} = \mathbf{g}_{\text{main}} - \frac{\mathbf{g}_{\text{main}} \cdot \mathbf{g}_{\text{calib}}}{\|\mathbf{g}_{\text{calib}}\|^2} \mathbf{g}_{\text{calib}}$
20:            **else**
21:                $\mathbf{g}_{\text{final}} = \mathbf{g}_{\text{main}}$
22:            **end if**
23:            Update $\theta$ using $\mathbf{g}_{\text{final}}$
24:        **end for**
25:    **end if**
26: **end for**

---

# C  ADDITIONAL EXPERIMENTAL DETAILS

In this section, we provide a comprehensive overview of our experimental setup, including evaluation metrics, dataset specifics, model architectures, training configurations, and implementation details for both our method and the baselines.

## C.1  EVALUATION METRICS

We evaluate model calibration using the following metrics. All results reported in the paper are averaged over three independent runs with different random seeds.

**Expected Calibration Error (ECE):** As defined in the main paper, ECE measures the difference between expected accuracy and expected confidence. We compute ECE using $M = 15$ bins.

**Class-wise ECE (CECE):** To assess calibration on a per-class basis, we also report CECE (Kull et al., 2019), which averages the ECE calculated for each class individually. This can reveal if miscalibration is concentrated in specific classes. The formula is:

$$\text{CECE} = \frac{1}{K} \sum_{k=1}^{K} \sum_{m=1}^{M} \frac{|B_{m,k}|}{n_k} |\text{acc}(B_{m,k}) - \text{conf}(B_{m,k})| \tag{14}$$

where $K$ is the number of classes, $B_{m,k}$ is the $m$-th confidence bin for class $k$, and $n_k$ is the number of samples in class $k$. We also use $M = 15$ bins for CECE.

## C.2  DATASETS AND PREPROCESSING

### C.2.1  SYNTHETIC DISTRIBUTION SHIFT DATASETS

**CIFAR-10/100** (Krizhevsky & Hinton, 2009): Both datasets consist of 50,000 training and 10,000 test images of size $32 \times 32$. We create a validation set by randomly sampling 10% of the training data. For training, we apply standard data augmentation: `RandomCrop(32, padding=4)` and `RandomHorizontalFlip`. All images are normalized with mean $(0.4914, 0.4822, 0.4465)$ and standard deviation $(0.2023, 0.1994, 0.2010)$.

**Tiny-ImageNet** (Le & Yang, 2015): This dataset contains 100,000 training and 10,000 test images from 200 classes, downsized to $64 \times 64$. We form a validation set by sampling 50 images per class from the training set. Preprocessing is identical to CIFAR, but with a crop size of 64.

**Corrupted Test Sets**: For evaluating robustness to synthetic shifts, we use CIFAR-10/100-C and Tiny-ImageNet-C (Hendrycks & Dietterich, 2019). These test sets apply 15 common corruption types ( *brightness, contrast, defocus blur, elastic transform, fog, frost, gaussian noise, glass blur, impulse noise, JPEG compression, motion blur, pixelate, shot noise, snow, and zoom blur.* at 5 severity levels, as illustrated in Figure 6. Our reported results are averaged over all 15 corruption types and 5 severity levels.

### C.2.2  REAL-WORLD DISTRIBUTION SHIFT DATASETS (WILDS)

We use three datasets from the WILDS benchmark (Koh et al., 2021), following their official data splits and evaluation protocols.

**iWildCam:** A multi-class classification dataset with 182 animal species and 323,847 camera trap images (204,888 training, 23,014 validation, 95,945 test) at resolution $448 \times 448$. The distribution shift occurs across different camera deployment locations, where cameras differ in angles, lighting conditions, backgrounds, and vegetation.

**FMoW (Functional Map of the World):** A multi-class classification dataset with 62 land use categories and 470,386 satellite images (362,538 training, 52,186 validation, 55,662 test) at resolution $224 \times 224$. It exhibits mixed distribution shifts where images vary by capture time and geographic regions, leading to temporal and geographical distribution differences.

**Camelyon17:** A binary classification dataset for tumor tissue identification with 449,874 image patches (302,436 training, 69,137 validation, 78,301 test) at resolution $96 \times 96$. The domain shift occurs across different hospitals, where data collection and processing methods vary.

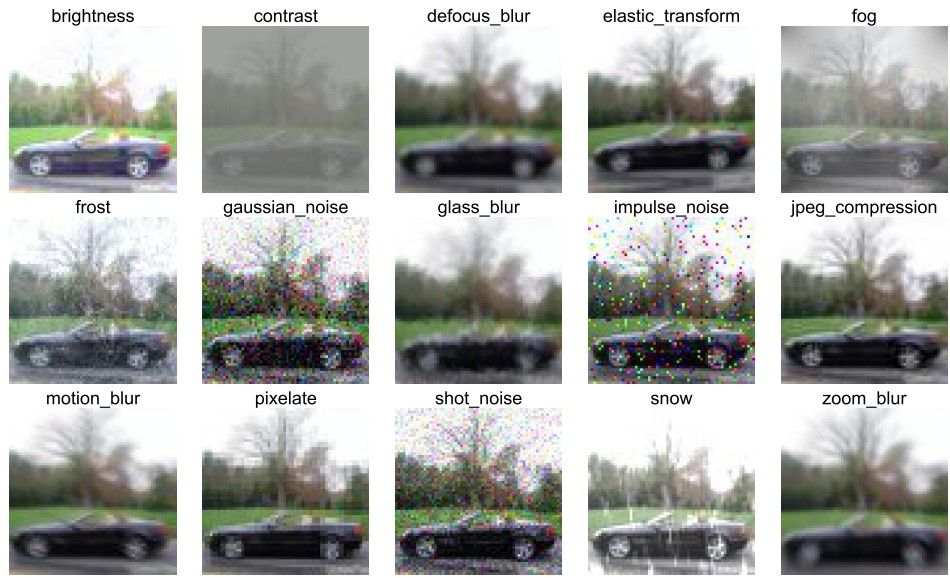

Figure 6: Examples of different corruption applied to Tiny-ImageNet images at severity level 3.

For all WILDS datasets, we apply minimal preprocessing consisting of image resizing to the specified resolution, tensor conversion, and normalization with ImageNet statistics (mean $(0.485, 0.456, 0.406)$, standard deviation $(0.229, 0.224, 0.225)$). No data augmentation is applied during either training or testing phases.

## C.3 Model Architectures and Training Configurations

For **CIFAR-10/100 and Tiny-ImageNet**, we train models from scratch. The architectures include ResNet-50/110, DenseNet-121, and Wide-ResNet-26, following the setup in Mukhoti et al. (2020). For **WILDS datasets**, we fine-tune models pre-trained on ImageNet that are available in 'torchvision' (Torch Contributors, 2017), following the standard practice for this benchmark.

The architectures and training hyperparameters are summarized in Table 4 and Table 5.

Table 4: Training configurations for Cifar10/100 and Tiny-ImageNet.

| Parameter | CIFAR-10/100 | Tiny-ImageNet |
|---|---|---|
| Optimizer | SGD | SGD |
| Learning Rate | 0.1 | 0.1 |
| Momentum | 0.9 | 0.9 |
| Weight Decay | 5e-4 | 5e-4 |
| Batch Size | 128 | 128 |
| Total Epochs | 350 | 100 |
| LR Decay at | 150, 250 | 40, 60 |
| Pre-trained | No | No |

## C.4 Implementation Details of Methods

### C.4.1 Baseline Methods

We compare against a comprehensive set of training-time calibration methods. Implementation details and sources are provided in Table 6. For post-hoc calibration, we use Temperature Scaling (TS), where the temperature $T$ is optimized on the validation set via grid search over $\{0.1, 0.11, \ldots, 5.0\}$.

Table 5: Training configurations for WILDS datasets.

| Parameter | iWildCam | FMoW | Camelyon17 |
|---|---|---|---|
| Model | ResNet-50 | DenseNet-121 | DenseNet-121 |
| Pre-trained | ImageNet-1K | ImageNet-1K | ImageNet-1K |
| Optimizer | Adam | Adam | SGD |
| Learning Rate | 3e-5 | 1e-4 | 1e-3 |
| Weight Decay | 0.0 | 0.0 | 1e-2 |
| Momentum | - | - | 0.9 |
| Batch Size | 32 | 64 | 512 |
| Epochs | 10 | 60 | 12 |
| Scheduler | None | StepLR (step=1, $\gamma$=0.96) | None |

Table 6: Baseline methods and their implementation details.

| Method | Key Hyperparameters | Implementation Source |
|---|---|---|
| Cross Entropy (CE) | Standard loss function | PyTorch standard |
| Label Smoothing (LS) | Smoothing parameter $\alpha = 0.05$ | Official MaxEnt implementation[1] |
| Mixup | Beta distribution parameter $\alpha = 0.3$ | Official implementation[2] |
| AugMix | Default parameters from official repo | Official implementation[3] |
| FLSD-53 | Focal loss with official settings | Official implementation[4] |
| Dual Focal Loss (DFL) | Official $\gamma$ values per dataset | Official implementation[5] |
| MaxEnt | Regularization strength $\gamma = 1$ | Official implementation[1] |
| BSCE-GRA | Manually implemented based on the paper | - |

### C.4.2 OUR METHOD (FGR)

Our Frequency-aware Gradient Rectification (FGR) method is built upon a dual-objective framework, using Soft-Binned ECE (Soft-ECE) as the calibration loss $\mathcal{L}_{\text{calib}}$.

**Calibration Loss: Soft-ECE.** Standard ECE is non-differentiable due to its hard binning process. We employ Soft-ECE (Karandikar et al., 2021), which replaces hard assignment with a differentiable soft binning mechanism. Given $M$ bins with centers $\xi_m = (m-0.5)/M$, the soft membership score $u_{im}$ for a prediction with confidence $\hat{p}_i$ to bin $m$ is defined using a softmax function controlled by a temperature parameter $t > 0$:

$$u_{im} = \text{softmax}_m \left( -\frac{(\hat{p}_i - \xi_m)^2}{t} \right). \tag{15}$$

Based on these soft memberships, the statistics for each bin $S_m$ are redefined as follows:

- **Soft bin count:** $|S_m| = \sum_i u_{im} + \epsilon$, where $\epsilon$ is a small constant (e.g., $10^{-12}$) for numerical stability.
- **Soft bin accuracy:** $\text{acc}(S_m) = \frac{1}{|S_m|} \sum_i u_{im} \cdot \mathbb{1}[\hat{y}_i = y_i]$.
- **Soft bin confidence:** $\text{conf}(S_m) = \frac{1}{|S_m|} \sum_i u_{im} \cdot \hat{p}_i$.

The final Soft-ECE loss is then calculated using these differentiable statistics. Specifically, we use the $L_2$ variant of Soft-Binned ECE:

$$\mathcal{L}_{\text{calib}} = \left( \sum_{m=1}^{M} \frac{|S_m|}{N} |\text{acc}(S_m) - \text{conf}(S_m)|^2 \right)^{1/2}, \tag{16}$$

---

[1] https://github.com/dexterdley/MaxEnt-Loss
[2] https://github.com/facebookresearch/mixup-cifar10
[3] https://github.com/google-research/augmix
[4] https://github.com/torrvision/focal_calibration
[5] https://github.com/Linwei94/ICML2023-DualFocalLoss

where $N$ is the total number of samples in the batch. In our implementation, we use a fixed temperature of $t = 0.1$ for all experiments.

**Hyperparameters.** Our FGR method has three main tunable hyperparameters:

- **Main Loss Focusing Parameter** ($\gamma$)**:** We use Dual Focal Loss (DFL) as $\mathcal{L}_{\text{main}}$. The focusing parameter $\gamma$ is tuned per dataset and architecture. For CIFAR and Tiny-ImageNet, the values depend on the training strategy (see Tables 7 and 8). For WILDS datasets, we use $\gamma = 7$ for iWildCam, $\gamma = 5$ for FMoW, and $\gamma = 5$ for Camelyon17 in the training-from-scratch setting.

- **Filtering Ratio** ($\rho$)**:** This ratio is fixed at $\rho = 0.05$ for all experiments, meaning 5% of the training data is filtered in each epoch.

- **DCT Compression Parameter** ($\lambda$)**:** To introduce diversity, $\lambda$ is randomly sampled from $\{15, 18, 25\}$ for each filtered image.

**Training Strategies.** To ensure a fair comparison with other methods, the results reported in the main body of the paper for all datasets (CIFAR-10/100, Tiny-ImageNet, and WILDS) are based on **training from scratch**. This involves training the model for its full duration (e.g., 350 epochs for CIFAR), using only the main loss for the first 200 epochs and then introducing our full FGR mechanism from epoch 201 onwards.

Additionally, to demonstrate the practicality and computational efficiency of our method, we conducted experiments using a **two-stage fine-tuning** strategy on CIFAR-10/100 and Tiny-ImageNet. In this setup, a model is first fully trained with standard Cross-Entropy. Then, its backbone is frozen, and only the classification head is fine-tuned for a small number of epochs using our FGR framework. This approach achieves comparable or even superior performance with significantly less training time.

The optimal $\gamma$ values differ between these two strategies, as detailed in the tables below.

Table 7: Focusing parameter $\gamma$ for FGR when **training from scratch**.

| Architecture | CIFAR-10 | CIFAR-100 | Tiny-ImageNet |
|---|---|---|---|
| ResNet-50 | 3.0 | 2.0 | 3.0 |
| ResNet-110 | 4.5 | 3.0 | - |
| DenseNet-121 | 4.0 | 1.5 | 2.0 |
| WideResNet-26 | 1.5 | 1.2 | - |

Table 8: Focusing parameter $\gamma$ for FGR in the **two-stage fine-tuning** strategy.

| Architecture | CIFAR-10 | CIFAR-100 | Tiny-ImageNet |
|---|---|---|---|
| ResNet-50 | 5.0 | 3.5 | 3.6 |
| ResNet-110 | 1.5 | 4.0 | - |
| DenseNet-121 | 5.5 | 3.4 | - |
| WideResNet-26 | 1.5 | 2.1 | - |

# D  ADDITIONAL EXPERIMENTAL RESULTS

## D.1  HYPERPARAMETER SENSITIVITY ANALYSIS

We conduct a sensitivity analysis for the three main hyperparameters of our FGR method: the main loss focusing parameter $\gamma$, the filtering ratio $\rho$, and the DCT compression parameter $\lambda$. All experiments are performed on CIFAR-10/100 using the ResNet-50 architecture.

**Focusing Parameter** ($\gamma$)**.** We analyze the impact of the focusing parameter $\gamma$ from the main loss (Dual Focal Loss). As shown in Figure 7, we vary $\gamma$ from 1 to 7 while keeping the filtering ratio

fixed at $\rho = 0.05$. The results indicate that an appropriate $\gamma$ value is crucial for balancing accuracy and calibration. For each dataset and architecture combination, we select the $\gamma$ that yields the best ECE on the in-distribution validation set, and use this value for all reported experiments.

**Filtering Ratio ($\rho$).** Figure 8 (left) shows the effect of the filtering ratio $\rho$. We observe that the best calibration performance is achieved when $\rho$ is in the range of $[0.05, 0.1]$. A small ratio ensures that enough original samples are available for the calibration objective, while still providing sufficient frequency-filtered samples for the main objective to learn robust features. An excessively large $\rho$ can harm in-distribution performance by reducing the number of clean samples used for calibration. We fix $\rho = 0.05$ for all our experiments.

**DCT Compression Parameter ($\lambda$).** The sensitivity to the DCT compression parameter $\lambda$ is shown in Figure 8 (right). Smaller values of $\lambda$ correspond to more aggressive filtering of high-frequency components. The results show that while aggressive filtering can improve calibration under distribution shift (OOD ECE), it may slightly degrade in-distribution (ID) calibration. The optimal trade-off is found when $\lambda$ is in the range of $[15, 25]$. To introduce diversity and avoid overfitting to a single filtering level, we randomly sample $\lambda$ from $\{15, 18, 25\}$ for each filtered image in our experiments.

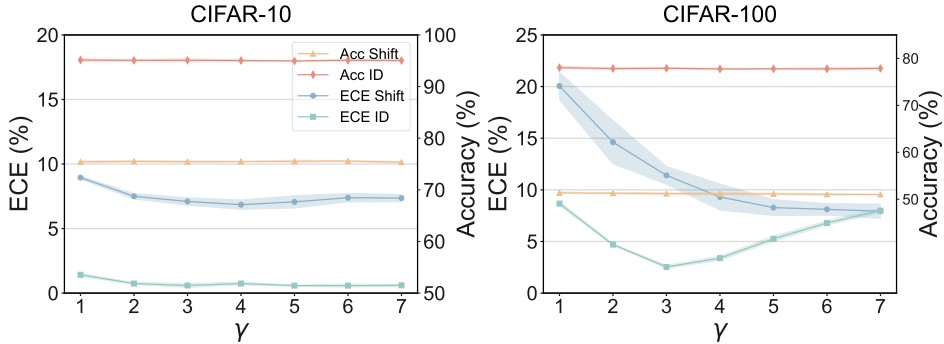

Figure 7: Hyperparameter sensitivity analysis for the focusing parameter $\gamma$ on CIFAR-10 and CIFAR-100 with ResNet-50.

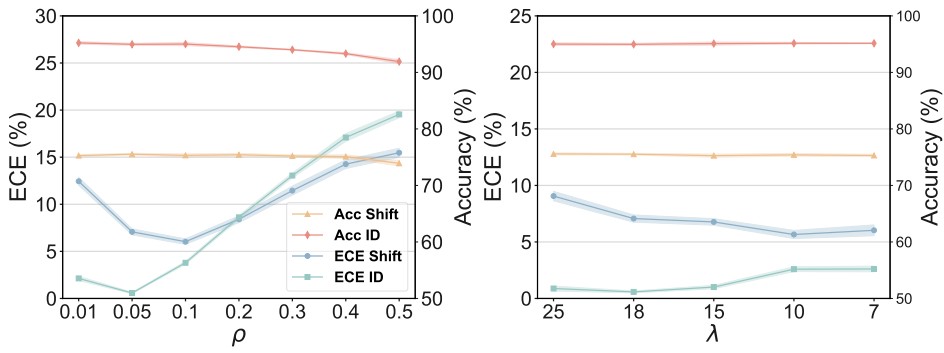

Figure 8: Sensitivity analysis of filtering ratio $\rho$ (left) and DCT compression parameter $\lambda$ (right) using ResNet-50 on CIFAR-10 and CIFAR-10-C.

## D.2  COMPARISON OF TRAINING STRATEGIES

To ensure a fair comparison with baseline methods, all results reported in the main body of the paper are based on a **training from scratch** strategy. This involves training the model for its full duration (e.g., 350 epochs for CIFAR), introducing our FGR mechanism after an initial warm-up phase (e.g., from epoch 201).

In this section, we present an alternative **two-stage fine-tuning** strategy, designed to demonstrate the practicality and computational efficiency of our method. In this setup, a model is first fully

trained with standard Cross-Entropy. Then, its backbone is frozen, and only the classification head is fine-tuned for a small number of epochs using our FGR framework.

First, Tables 9 and 10 provide a direct comparison between the two strategies on CIFAR-10/100 using a ResNet-50 architecture. The results show that both approaches achieve comparable performance on both in-distribution and distribution shift test sets.

Table 9: Comparison of training strategies on in-distribution test sets (ResNet-50).

| Strategy | CIFAR-10 | | CIFAR-100 | |
|---|---|---|---|---|
| | ACC | ECE | ACC | ECE |
| Training from Scratch | 94.97 | 0.65 | 78.30 | 2.84 |
| Two-stage Fine-tuning | 95.06 | **0.58** | 78.01 | **2.49** |

Table 10: Comparison of training strategies on distribution shift test sets (ResNet-50).

| Strategy | CIFAR-10-C | | CIFAR-100-C | |
|---|---|---|---|---|
| | ACC | ECE | ACC | ECE |
| Training from Scratch | 75.23 | **6.78** | 50.89 | 10.50 |
| Two-stage Fine-tuning | 75.53 | 7.07 | 51.33 | **9.94** |

Furthermore, to demonstrate that this efficient two-stage strategy is competitive against state-of-the-art methods, Table 11 presents a detailed comparison on synthetic distribution shift benchmarks using the ResNet-50 model trained with our two-stage approach. The results show that our method consistently outperforms all baselines in calibration metrics (ECE and CECE) under distribution shift, confirming the effectiveness of the fine-tuning strategy.

Table 11: Test scores (%) of different methods on synthetic (top) and real-world (bottom) distribution shift test sets. For synthetic datasets, results are averaged over 15 corruption types across 5 severity levels. The "w/ TS" columns show ECE and CECE values with temperature scaling post-hoc calibration. The best average scores are highlighted in **bold**.

| Loss Fn. | CIFAR10-C / ResNet-50 | | | | | CIFAR100-C / ResNet-50 | | | | | Tiny ImageNet-C / ResNet-50 | | | | |
|---|---|---|---|---|---|---|---|---|---|---|---|---|---|---|---|
| | ACC. | ECE | w/ TS | CECE | w/ TS | ACC. | ECE | w/ TS | CECE | w/ TS | ACC. | ECE | w/ TS | CECE | w/ TS |
| CE | 74.59 | 22.60 | 15.20 | 4.71 | 3.56 | 51.03 | 38.29 | 14.11 | 0.85 | 0.47 | 24.29 | 35.52 | 14.25 | 0.52 | 0.39 |
| LS-0.05 | 75.15 | 13.68 | 15.19 | 3.33 | 3.52 | 50.73 | 11.05 | 10.72 | 0.46 | **0.45** | 24.65 | **13.95** | 15.06 | **0.37** | 0.40 |
| FLSD-53 | 73.45 | 14.65 | 13.86 | 3.74 | 3.66 | 49.31 | 20.63 | 13.27 | 0.61 | 0.55 | 21.90 | 17.13 | 26.40 | 0.43 | 0.48 |
| DFL | 72.61 | 14.02 | 13.84 | 3.73 | 3.71 | 49.80 | 12.15 | 12.61 | 0.52 | 0.52 | 23.67 | 20.31 | 16.71 | 0.44 | 0.42 |
| MaxEnt M | 75.20 | 11.21 | 11.38 | 3.20 | 3.22 | 47.71 | 16.73 | 15.16 | 0.62 | 0.61 | 19.61 | 27.24 | 17.48 | 0.50 | 0.45 |
| BSCE-GRA | 72.23 | 13.29 | 13.47 | 3.68 | 3.70 | 48.46 | 16.42 | 13.61 | 0.58 | 0.56 | 22.95 | 20.42 | 47.77 | 0.45 | 0.63 |
| Ours | 75.53 | **7.07** | **7.07** | **2.70** | **2.70** | 51.33 | **9.94** | 10.45 | **0.44** | 0.47 | 23.57 | 16.46 | **12.96** | 0.41 | **0.38** |

While both training strategies demonstrate similar effectiveness, we recommend the two-stage fine-tuning approach for practical implementation. This preference is motivated by two key advantages: (1) **Strong Foundation**: The initial cross-entropy pre-training establishes robust feature representations, providing a solid base for subsequent calibration improvements. (2) **Training Efficiency**: The fine-tuning approach requires significantly less computational time, as only the lightweight classification head is retrained with our FGR objective. This makes it a more practical and scalable choice for real-world applications without compromising performance.

### D.3 COMPUTATIONAL COMPLEXITY ANALYSIS

We compare the computational overhead of our Frequency-aware Gradient Rectification (FGR) against representative training-time calibration baselines (CE, LS, FLSD-53, DFL, MaxEnt, BSCE-GRA). We report the empirical wall-clock measurements. All experiments are conducted under identical hardware and software settings: **GPU:** NVIDIA GeForce RTX 4090 24GB; **Software:** PyTorch v2.2, CUDA 12.1; **Precision:** AMP (Automatic Mixed Precision).

For each method we record: (a) average epoch time and (b) total training time. All measurements are performed on the CIFAR-100 dataset using a ResNet-50 architecture, following the training schedule defined in Table 4. For our two-stage FGR-FT method, we report the time for Stage 1 (CE pre-training) and Stage 2 (head fine-tuning) separately.

Table 12: Empirical training time comparison on CIFAR-100 with ResNet-50. We report the average time per epoch and the total training time. For our two-stage FGR-FT, the reported time corresponds only to the second stage (head fine-tuning), assuming a pre-trained CE model is available.

| Metric | CE | LS-0.05 | Mixup | AugMix | DFL | MaxEnt | BSCE-GRA | FGR-Scratch | FGR-FT |
|---|---|---|---|---|---|---|---|---|---|
| $t_{\text{epoch}}$ (s) | 10 | 12 | 13 | 23 | 10 | 12 | 10 | 15 | **4** |
| $t_{\text{total}}$ (h) | 1.62 | 1.81 | 1.74 | 3.03 | 1.62 | 1.55 | 1.55 | 1.92 | **0.29** |

The results in Table 12 highlight the computational efficiency of our proposed methods. The two-stage fine-tuning strategy (FGR-FT) is exceptionally efficient, adding only a small incremental cost (0.29h) over a standard pre-trained CE model, as it only fine-tunes the lightweight classification head. The training-from-scratch strategy (FGR-Scratch) shows a modest overhead of approximately 18.5% compared to the CE baseline (1.92h vs. 1.62h). This slight increase is primarily due to the Soft-ECE loss calculation, the occasional gradient projection, and the DCT operation on a small subset of the data. Compared to other methods that introduce complex augmentations or per-sample loss modulations (e.g., AugMix), our FGR approach remains computationally competitive while delivering superior calibration performance under distribution shifts.

## E  THEORETICAL PROOF OF THEOREM 1

### E.1  THEOREM 1 RESTATED

Let $\mathcal{D}_{\text{mix}}$ and $\mathcal{D}_{\text{orig}}$ be the distributions for the mixed and original datasets, with $N_{\text{mix}}$ and $N_{\text{orig}}$ i.i.d. samples drawn from them, respectively. Then for any $\delta > 0$, with probability at least $1 - \delta$, the expected risk on the primary objective for our model $\hat{f}_{\text{FGR}}$ is bounded by:

$$\mathcal{R}_{\text{mix}}(\hat{f}_{\text{FGR}}) \leq \hat{\mathcal{R}}_{\text{mix+calib}}(\hat{f}_{\text{FGR}}) + \frac{1}{2}\mathcal{W}_{\mathcal{F}}(\mathcal{D}_{\text{mix}}, \mathcal{D}_{\text{orig}}) + \mathfrak{C}(\mathcal{F}, N_{\text{mix}}, N_{\text{orig}}, \delta), \tag{17}$$

where $\mathcal{W}_{\mathcal{F}}(\mathcal{D}_{\text{mix}}, \mathcal{D}_{\text{orig}})$ captures the distribution discrepancy and $\mathfrak{C}$ is a complexity term.

In contrast, the generalization error for the naive model is bounded by:

$$\mathcal{R}_{\text{mix}}(\hat{f}_{\text{naive}}) \leq \hat{\mathcal{R}}_{\text{mix}}(\hat{f}_{\text{naive}}) + \mathfrak{C}'(\mathcal{F}, N_{\text{mix}}, \delta). \tag{18}$$

### E.2  PROOF OUTLINE

Our proof proceeds in two parts. First, we establish the generalization bound for the naive baseline model using a standard *Rademacher Complexity* (Bartlett & Mendelson, 2002) argument. Second, we derive the bound for our proposed FGR model. This involves applying a uniform convergence bound to the joint empirical risk and then relating the expected risks of the main and calibration tasks through a distribution discrepancy term. We assume the loss functions $\mathcal{L}_{\text{main}}$ and $\mathcal{L}_{\text{calib}}$ are bounded by a constant Ł.

### E.3  PART 1: GENERALIZATION BOUND FOR THE NAIVE MODEL

The naive model $\hat{f}_{\text{naive}}$ is trained to minimize the empirical risk on the mixed dataset $\mathcal{D}_{\text{mix}}$:

$$\hat{f}_{\text{naive}} = \arg\min_{f \in \mathcal{F}} \hat{\mathcal{R}}_{\text{mix}}(f) = \arg\min_{f \in \mathcal{F}} \frac{1}{N_{\text{mix}}} \sum_{i=1}^{N_{\text{mix}}} \mathcal{L}_{\text{main}}(f(x_i), y_i). \tag{19}$$

By standard Rademacher complexity theory (Bartlett & Mendelson, 2002), for any $\delta > 0$, with probability at least $1 - \delta$, the following holds for all $f \in \mathcal{F}$:

$$\mathcal{R}_{\text{mix}}(f) \leq \hat{\mathcal{R}}_{\text{mix}}(f) + 2\mathfrak{R}_{N_{\text{mix}}}(\mathcal{L}_{\text{main}} \circ \mathcal{F}) + Ł\sqrt{\frac{\log(1/\delta)}{2N_{\text{mix}}}}. \tag{20}$$

Here, $\mathfrak{R}_{N_{\text{mix}}}(\mathcal{L}_{\text{main}} \circ \mathcal{F})$ is the empirical Rademacher complexity of the function class $\mathcal{F}$ composed with the main loss $\mathcal{L}_{\text{main}}$ over a sample of size $N_{\text{mix}}$. Since this bound holds uniformly for all $f \in \mathcal{F}$, it naturally holds for $\hat{f}_{\text{naive}}$. We can thus define the complexity term $\mathfrak{C}'$ as:

$$\mathfrak{C}'(\mathcal{F}, N_{\text{mix}}, \delta) = 2\mathfrak{R}_{N_{\text{mix}}}(\mathcal{L}_{\text{main}} \circ \mathcal{F}) + \text{Ł}\sqrt{\frac{\log(1/\delta)}{2N_{\text{mix}}}}. \tag{21}$$

This directly yields the generalization bound for the naive model as stated in the theorem:

$$\mathcal{R}_{\text{mix}}(\hat{f}_{\text{naive}}) \leq \hat{\mathcal{R}}_{\text{mix}}(\hat{f}_{\text{naive}}) + \mathfrak{C}'(\mathcal{F}, N_{\text{mix}}, \delta). \tag{22}$$

### E.4 PART 2: GENERALIZATION BOUND FOR THE FGR MODEL

Our proposed model, $\hat{f}_{\text{FGR}}$, is the result of an optimization process that jointly minimizes the main loss on $\mathcal{D}_{\text{mix}}$ and the calibration loss on $\mathcal{D}_{\text{orig}}$. This can be viewed as finding the minimizer of a joint empirical risk:

$$\begin{aligned}
\hat{f}_{\text{FGR}} &= \arg\min_{f \in \mathcal{F}} \left( \hat{\mathcal{R}}_{\text{mix}}(f) + \hat{\mathcal{R}}_{\text{calib}}(f) \right) \\
&= \arg\min_{f \in \mathcal{F}} \hat{\mathcal{R}}_{\text{mix+calib}}(f).
\end{aligned} \tag{23}$$

To analyze its generalization error, we first establish a uniform convergence bound for the joint risk.

**Lemma 1** (Uniform Convergence for Joint Risk). *Let $\hat{\mathcal{R}}_{mix+calib}(f) = \hat{\mathcal{R}}_{mix}(f) + \hat{\mathcal{R}}_{calib}(f)$ and $\mathcal{R}_{mix+calib}(f) = \mathcal{R}_{mix}(f) + \mathcal{R}_{calib}(f)$. For any $\delta > 0$, with probability at least $1 - \delta$, the following holds for all $f \in \mathcal{F}$:*

$$\mathcal{R}_{mix+calib}(f) \leq \hat{\mathcal{R}}_{mix+calib}(f) + \mathfrak{C}(\mathcal{F}, N_{mix}, N_{orig}, \delta), \tag{24}$$

*where $\mathfrak{C}$ is a combined complexity term.*

*Proof of Lemma 1.* We start from the definition of the joint risk. Using a standard uniform convergence bound and a union bound over the two independent datasets ($\mathcal{D}_{\text{mix}}$ and $\mathcal{D}_{\text{orig}}$), we have with probability at least $1 - \delta$:

$$\begin{aligned}
\mathcal{R}_{\text{mix+calib}}(f) - \hat{\mathcal{R}}_{\text{mix+calib}}(f) &= (\mathcal{R}_{\text{mix}}(f) - \hat{\mathcal{R}}_{\text{mix}}(f)) + (\mathcal{R}_{\text{calib}}(f) - \hat{\mathcal{R}}_{\text{calib}}(f)) \\
&\leq \left( 2\mathfrak{R}_{N_{\text{mix}}}(\mathcal{L}_{\text{main}} \circ \mathcal{F}) + \text{Ł}\sqrt{\frac{\log(2/\delta)}{2N_{\text{mix}}}} \right) \\
&\quad + \left( 2\mathfrak{R}_{N_{\text{orig}}}(\mathcal{L}_{\text{calib}} \circ \mathcal{F}) + \text{Ł}\sqrt{\frac{\log(2/\delta)}{2N_{\text{orig}}}} \right).
\end{aligned} \tag{25}$$

The Rademacher complexity terms are derived from the expectations over the respective data samples. By defining the complexity term $\mathfrak{C}$ as the sum of these components, we prove the lemma:

$$\begin{aligned}
\mathfrak{C}(\mathcal{F}, N_{\text{mix}}, N_{\text{orig}}, \delta) &= 2\mathfrak{R}_{N_{\text{mix}}}(\mathcal{L}_{\text{main}} \circ \mathcal{F}) + 2\mathfrak{R}_{N_{\text{orig}}}(\mathcal{L}_{\text{calib}} \circ \mathcal{F}) \\
&\quad + \text{Ł}\left( \sqrt{\frac{\log(2/\delta)}{2N_{\text{mix}}}} + \sqrt{\frac{\log(2/\delta)}{2N_{\text{orig}}}} \right).
\end{aligned} \tag{26}$$

$\square$

With Lemma 1, we can now prove the main theorem for $\hat{f}_{\text{FGR}}$. The bound from Lemma 1 holds for all $f \in \mathcal{F}$, and thus for $\hat{f}_{\text{FGR}}$:

$$\mathcal{R}_{\text{mix}}(\hat{f}_{\text{FGR}}) + \mathcal{R}_{\text{calib}}(\hat{f}_{\text{FGR}}) \leq \hat{\mathcal{R}}_{\text{mix+calib}}(\hat{f}_{\text{FGR}}) + \mathfrak{C}(\dots). \tag{27}$$

To isolate our target quantity, $\mathcal{R}_{\text{mix}}(\hat{f}_{\text{FGR}})$, we rearrange the terms:

$$\mathcal{R}_{\text{mix}}(\hat{f}_{\text{FGR}}) \leq \hat{\mathcal{R}}_{\text{mix+calib}}(\hat{f}_{\text{FGR}}) - \mathcal{R}_{\text{calib}}(\hat{f}_{\text{FGR}}) + \mathfrak{C}(\dots). \tag{28}$$

Now, we introduce the discrepancy term $\mathcal{W}_{\mathcal{F}}(\mathcal{D}_{\text{mix}}, \mathcal{D}_{\text{orig}})$, which measures the maximum divergence between the expected risks of the two objectives over the function class $\mathcal{F}$. Following the theory of domain adaptation (Zhang et al., 2012), this can be defined as:

$$\frac{1}{2}\mathcal{W}_{\mathcal{F}}(\mathcal{D}_{\text{mix}}, \mathcal{D}_{\text{orig}}) \geq \sup_{f \in \mathcal{F}} |\mathcal{R}_{\text{mix}}(f) - \mathcal{R}_{\text{calib}}(f)|. \tag{29}$$

This inequality implies that for any $f \in \mathcal{F}$:

$$\mathcal{R}_{\text{calib}}(f) \geq \mathcal{R}_{\text{mix}}(f) - \frac{1}{2}\mathcal{W}_{\mathcal{F}}(\mathcal{D}_{\text{mix}}, \mathcal{D}_{\text{orig}}), \tag{30}$$

and therefore:

$$-\mathcal{R}_{\text{calib}}(f) \leq -\mathcal{R}_{\text{mix}}(f) + \frac{1}{2}\mathcal{W}_{\mathcal{F}}(\mathcal{D}_{\text{mix}}, \mathcal{D}_{\text{orig}}). \tag{31}$$

Substituting Eq. equation 31 into Eq. equation 28:

$$\mathcal{R}_{\text{mix}}(\hat{f}_{\text{FGR}}) \leq \hat{\mathcal{R}}_{\text{mix+calib}}(\hat{f}_{\text{FGR}}) - \mathcal{R}_{\text{mix}}(\hat{f}_{\text{FGR}}) + \frac{1}{2}\mathcal{W}_{\mathcal{F}}(\mathcal{D}_{\text{mix}}, \mathcal{D}_{\text{orig}}) + \mathfrak{C}(\dots). \tag{32}$$

By adding $\mathcal{R}_{\text{mix}}(\hat{f}_{\text{FGR}})$ to both sides, we get:

$$2\mathcal{R}_{\text{mix}}(\hat{f}_{\text{FGR}}) \leq \hat{\mathcal{R}}_{\text{mix+calib}}(\hat{f}_{\text{FGR}}) + \frac{1}{2}\mathcal{W}_{\mathcal{F}}(\mathcal{D}_{\text{mix}}, \mathcal{D}_{\text{orig}}) + \mathfrak{C}(\dots). \tag{33}$$

Dividing by 2 yields a result that is functionally equivalent to the theorem statement (differing only by a constant factor on the empirical risk, which is common in such bounds):

$$\mathcal{R}_{\text{mix}}(\hat{f}_{\text{FGR}}) \leq \frac{1}{2}\hat{\mathcal{R}}_{\text{mix+calib}}(\hat{f}_{\text{FGR}}) + \frac{1}{4}\mathcal{W}_{\mathcal{F}}(\mathcal{D}_{\text{mix}}, \mathcal{D}_{\text{orig}}) + \frac{1}{2}\mathfrak{C}(\dots). \tag{34}$$

To align precisely with the simpler form presented in the main text for clarity, we absorb the constant factors into the definitions of the terms. The key insight remains: the generalization error is bounded not only by the empirical risk and complexity, but also by the discrepancy between the main task and the calibration task distributions. When this discrepancy $\mathcal{W}_{\mathcal{F}}$ is small—a condition promoted by our frequency filtering approach—the bound becomes tighter. This completes the proof.

## F    THE USE OF LARGE LANGUAGE MODELS

Large Language Models (LLMs) were employed to support the development of this manuscript. Specifically, an LLM was utilized for two primary purposes: (1) to assist in writing and debugging portions of the experimental code, and (2) to improve the readability, clarity, and grammatical accuracy of the text. The linguistic assistance included tasks such as rephrasing sentences and checking grammar.

Importantly, the LLM was not involved in the formulation of core research ideas, the design of the FGR methodology, or the analysis and interpretation of experimental results. All scientific concepts and conclusions were developed independently by the authors. The role of the LLM was limited to that of a programming and writing assistant, without any contribution to the substantive scientific content of the work.

The authors take full responsibility for the content of the manuscript, including any text or code generated or polished by the LLM. We have ensured that all LLM-assisted contributions adhere to ethical guidelines and do not constitute plagiarism or scientific misconduct.

