# OpenReview forum: "Gradient Rectification for Robust Calibration under Distribution Shift"
_ICLR.cc/2026/Conference — ICLR 2026 Conference Withdrawn Submission_

### Official Review · Reviewer_tDTp · 2025-10-16

**Soundness:** 3
**Presentation:** 3
**Contribution:** 2
**Rating:** 4
**Confidence:** 3

**Summary:**

This paper proposes a novel training loss design aimed at improving calibration under dataset shift. The proposed method, Frequency-aware Gradient Rectification (FGR), is based on the assumption that dataset shift mainly occurs in the high-frequency components of images. It applies low-pass filtering to reduce feature representations that are likely to be spurious, and additionally proposes an orthogonal projection of the original loss onto the calibration loss to prevent the degradation of in-distribution (ID) confidence caused by filtering. Through experiments, the method demonstrates comparable or superior performance to existing methods on ID data in terms of ECE and CECE, while also showing improvements in multiple out-of-distribution (OOD) cases.

**Strengths:**

- Clarity of presentation: The main argument and logical flow of the paper are easy to follow and clearly articulated throughout.
- The proposed method does not require access to the target distribution: This approach overcomes a major limitation of conventional methods that rely on access to the target distribution.

**Weaknesses:**

- Limitations of frequency-based assumptions in distribution shift: The method developed in this paper assumes that invariant features are primarily embedded in low-frequency components, whereas spurious features arise from high-frequency regions. However, this assumption does not consistently reflect real-world conditions. For example, distribution shifts may result from variations in low-frequency aspects such as overall image tone (e.g., brightness or hue), or from frequency-independent structural changes, including shifts in the spatial relationship between foreground and background. Furthermore, in fine-grained classification tasks, critical invariant features may inherently reside in high-frequency components. These scenarios reveal a key limitation of the proposed method, which may underperform when such complex or non-frequency-aligned distribution shifts are present.


- Ambiguity regarding the novelty of gradient rectification: Gradient surgery [1] in multi-task learning is a well-known technique for resolving conflicts between gradients from different loss functions. The proposed gradient rectification is conceptually quite similar, and the paper does not clearly explain where its novelty lies or how it offers superior calibration under distribution shift.

- Limited evaluation metrics for calibration: Only ECE and CECE are used, both of which are known to be sensitive to confidence bias in the model. Evaluation using Adaptive Calibration Error (ACE) [2], which addresses this issue, would provide a more convincing assessment.

[1] Yu, Tianhe, et al. "Gradient surgery for multi-task learning." Advances in neural information processing systems 33 (2020): 5824-5836.

[2] Nixon, Jeremy, et al. "Measuring calibration in deep learning." CVPR workshops. Vol. 2. No. 7. 2019.

**Questions:**

See weaknesses

---

> ### Author Response · Authors · 2025-11-24
> **Author response to Reviewer tDTp - Part 1**
>
> Thank you for your comprehensive review. We greatly value your suggestions regarding the frequency domain assumption, novelty, and evaluation metrics. We respond to each point below.
>
> **W1: Limitations of frequency-based assumptions in distribution shift.**
>
> We fully agree that this is an effective and important limitation. We do not claim that all shifts are high-frequency, nor that all invariant features are low-frequency. Our goal is to mitigate the failure of overconfidence caused by frequency shortcut cues and improve calibration in scenarios without target access.
>
> To address your concerns, we present detailed **ECE** and **ACC** comparisons between standard Cross Entropy (CE) training and FGR across 15 corruptions on Cifar10-C. Since different corruptions affect different frequencies, accuracy variations may differ: FGR underperforms CE on 5/15 corruptions but significantly improves calibration across all 15 corruptions. This aligns with our design goal, demonstrating that FGR effectively suppresses “overconfidence” regardless of where shifts occur while preserving accuracy as much as possible. Additionally, in response to **reviewer AK1A's weakness 4**, we supplemented the performance of the proposed method on the semantically shifted Office-Home dataset. Results demonstrate that FGR provides consistent calibration gains even for data exhibiting semantic distribution shifts.
>
> (Due to width limitations, we have split each table into two rows.)
>
> | ACC  | brightness | fog        | defocus_blur | elastic_transform | snow       | zoom_blur  | contrast   | frost      | motion_blur |
> | ---- | ---------- | ---------- | ------------ | ----------------- | ---------- | ---------- | ---------- | ---------- | ----------- |
> | CE   | 92.80%     | 87.69%     | **83.35%**   | 82.73%            | 81.90%     | **78.69%** | **78.42%** | 78.31%     | 77.52%      |
> | FGR  | **93.46%** | **89.07%** | 81.45%       | **83.09%**        | **83.82%** | 76.49%     | 77.54%     | **81.68%** | **77.92%**  |
>
> | ACC  | jpeg_compression | pixelate   | shot_noise | impulse_noise | glass_blur | gaussian_noise |
> | ---- | ---------------- | ---------- | ---------- | ------------- | ---------- | -------------- |
> | CE   | 77.30%           | **75.22%** | 61.53%     | **60.78%**    | 53.72%     | 50.21%         |
> | FGR  | **79.56%**       | 73.42%     | **63.51%** | 53.90%        | **60.47%** | **52.26%**     |
>
> | ECE  | brightness  | fog         | defocus_blur | elastic_transform | snow        | zoom_blur   | contrast    | frost       | motion_blur |
> | ---- | ----------- | ----------- | ------------ | ----------------- | ----------- | ----------- | ----------- | ----------- | ----------- |
> | CE   | 0.02571     | 0.06027     | 0.08967      | 0.0934            | 0.103       | 0.12279     | 0.11196     | 0.12816     | 0.13624     |
> | FGR  | **0.00845** | **0.01837** | **0.05438**  | **0.05302**       | **0.03393** | **0.07526** | **0.07239** | **0.01892** | **0.07663** |
>
> | ECE  | jpeg_compression | pixelate    | shot_noise  | impulse_noise | glass_blur  | gaussian_noise |
> | ---- | ---------------- | ----------- | ----------- | ------------- | ----------- | -------------- |
> | CE   | 0.13344          | 0.14787     | 0.26151     | 0.24921       | 0.31159     | 0.35691        |
> | FGR  | **0.05753**      | **0.12341** | **0.08788** | **0.22918**   | **0.09566** | **0.13113**    |
>
>
> We will incorporate this discussion as an explicit limitation in the paper's conclusion section.

---

> ### Author Response · Authors · 2025-11-24
> **Author response to Reviewer tDTp - Part 2**
>
> **W2: Novelty of gradient rectification.**
>
> We acknowledge that our approach shares the same geometric intuition with Gradient Surgery (GS). However, the gradient correction we propose differs fundamentally from GS in both its optimization objective and mathematical formulation, since we are addressing an entirely different problem.
>
> 1. The optimization objectives is different: GS is used for multi-task learning, aiming to balance two equally weighted tasks ($L_A, L_B$). While FGR aims to solve a constrained optimization problem: minimizing $L_{main}$ subject to the constraint that $L_{calib}$ does not increase. Improvements in calibration performance should not come at the expense of accuracy.
> 2. The update mechanisms is different: In GS, when $g_A \cdot g_B < 0$, both gradients are modified. In FGR, due to the above-mentioned difference in optimization objectives, we employ an asymmetric update. If $g_{main} \cdot g_{calib} < 0$, only $g_{main}$ is modified. $g_{calib}$ is treated as the "constraint direction," and $g_{main}$ is projected onto the normal plane of $g_{calib}$. Backpropagation is ultimately performed based on $g_{main}$.
>
> In summary, our novelty lies in: (1) modeling the robust calibration problem as an asymmetric constrained optimization problem; (2) designing a parameter-free gradient projection specifically to solve this problem, specifically addressing the trade-off where "filtering leads to degraded ID calibration." We will add this explicit comparison in Section 4.2 of the main text.
>
> **W3: Add additional evaluation metrics.**
>
> Thank you for your constructive suggestions. We fully agree that the evaluation should be more comprehensive. We have re-run the evaluation on the saved checkpoints, incorporating the ACE metric. The new ACE results ( will be added to the main text tables and appendix) similarly confirm our findings, demonstrating that FGR consistently outperforms the baseline method on this more robust metric.
>
> We hope our response could address your concerns, and we also welcome any further feedback you might provide to enhance our manuscript.

---

> > ### Comment · Reviewer_tDTp · 2025-11-25
> > **Reply to authors**
> >
> > Thank you for the careful handling of my concerns.
> >
> > While W2 and W3 are resolved, questions remain for W1. I understand the method combines a low-pass filter with gradient rectification. For corruptions affecting different frequency bands, I can see how calibration-based gradient rectification improves ECE; however, the role of the low-pass filter in this setting is still unclear to me. An ablation that applies each component independently—filter-only and rectification-only—would clarify which part contributes to improvements, or whether both are necessary together.
> >
> > Intuitively, in realistic settings invariant features and shortcut cues can swap across frequencies in complex ways, which may weaken the utility of low-pass filtering. If so, the difference from existing calibration-only algorithms could be small. It would help to include theoretical or empirical comparison against calibration-focused approaches such as CLOvE by [3] to delineate the added value of frequency-aware filtering.
> >
> > [3] Wald, Yoav, et al. "On calibration and out-of-domain generalization." Advances in neural information processing systems 34 (2021): 2215-2227.

---

### Official Review · Reviewer_AK1A · 2025-10-26

**Soundness:** 2
**Presentation:** 3
**Contribution:** 2
**Rating:** 4
**Confidence:** 3

**Summary:**

This paper proposes Frequency-aware Gradient Rectification (FGR) to improve model calibration under distribution shift without accessing target-domain data. For details, FGR combines low-pass filtering to suppress spurious high-frequency features with gradient rectification to maintain in-distribution calibration by resolving gradient conflicts. Experiments achieves superior robustness and calibration compared to existing methods.

**Strengths:**

(1) Exploring domain shift calibration is important.

(2) The paper is well written and easy to follow

**Weaknesses:**

(1) Why use DCT is not clear. As described in the introduction, "learning to recognize 'birds' based on special texture (e.g., green
leafy patterns) rather than shape . Motivated by this, we apply Discrete Cosine Transform (DCT)
filtering to isolate low-frequency image components, encouraging the model to rely on shape-related
information that is more consistent across distributions.". However, i do not think this is well motivated to use DCT. More analysis are needed.

(2) Why use gradient rectification is also not clear.

(3)The paper lacks an explicit analysis or report of this computational overhead compared to standard training.


(4) The method’s robustness to semantic distribution shifts remains untested. For example. Office-Home dataset can be tested.

**Questions:**

my major concern is the relationship between the problem faced by the author (calibration) and the solution strategy is not clear.

---

> ### Author Response · Authors · 2025-11-24
> **Author response to Reviewer AK1A - Part 1**
>
> We sincerely thank you for your thorough review and acknowledge the importance of our work and the clarity of our writing. Below, we address each point raised and provide corresponding clarifications.
>
> **W1: Why use DCT.**
>
> Please allow us to clarify the rationale for employing low-pass filtering. Previous research indicates that neural networks tend to seek simple solutions for classification problems, often exploiting notable frequency features as spurious shortcuts [1]. When distributions shift, these shortcut features exhibit the most statistical variability, leading models to generate distorted high-confidence predictions based on unstable signals. Furthermore, many common distribution shifts (e.g., corruption in Cifar-C, Fig. 1) primarily alter high-frequency components of images [2].
>
> Therefore, by applying low-pass processing (DCT filtering) to a subset of training samples, we suppressed high-frequency shortcuts and encouraged the model to rely more on stable low-frequency cues. This approach achieved significant improvements in shift scenario calibration without accessing the target domain.
>
> Additionally, we adopt DCT due to its energy compression properties and block-level processing mechanism, which are also easy to implement. DCT's localized processing effectively avoids the global ringing artifacts common in Fast Fourier Transform (FFT), preventing distortion of image structure. In contrast, Gaussian Blur applies uniform smoothing across the entire image, leading to the loss of edge information that serves as crucial shape cues. Thus, DCT offers the optimal balance between suppressing spurious high-frequency features and preserving local semantic structure.
>
> **W2: Why use gradient rectification.**
>
> Following our response to W1, we wish to clarify that gradient rectification is not an independent module, but rather a mechanism necessarily introduced to resolve the calibration conflicts between the in-distribution and out-of-distribution data caused by DCT filtering. As stated in W1, while low-pass filtering effectively enhances OOD robustness, it also removes certain high-frequency details, causing the model to become less confident on clean ID data (as shown in Figure 3). Therefore, gradient rectification is exactly the parameter-free tool we designed to resolve this conflict. It ensures the model learns OOD calibration robustness without sacrificing ID calibration performance.
>
> **W3: Lack of computational overhead reporting.**
>
> We do recognize the importance of computational overhead analysis, but due to page constraints in the main text, we previously reported this analysis in Appendix D.3 (Table 12). The analysis indicates that FGR does not significantly increase computational overhead. On the CIFAR-100 dataset, our recommended FGR-FT (fine-tuning) strategy requires only 0.29 hours. FGR-Scratch (from-scratch training) incurs only about 18.5% overhead (1.92h vs 1.62h), which is highly competitive. We will summarize the core conclusions from Table 12 and move them to the main text's experimental section.
>
>
> References:
>
> [1] Wang et al., What do neural networks learn in image classification?  A frequency shortcut perspective. In Proceedings of the IEEE/CVF International Conference on Computer Vision, pp. 1433–1442, 2023.
>
> [2] Zhang et al., Adversarial Defense by Suppressing High-frequency Components. arXiv preprint arXiv:1908.06566, 2019.

---

> ### Author Response · Authors · 2025-11-24
> **Author response to Reviewer AK1A - Part 2**
>
> **W4: Robustness on the Office-Home Dataset.**
>
> Thanks, this is a highly valuable suggestion that significantly broadens the scope of our evaluation. We have conducted supplementary experiments on the Office-Home dataset and will incorporate them into the main text.
>
> Experimental details and results are as follows: We performed the full-funnel fine-tuning on all methods using a ResNet-50 pre-trained on ImageNet-1k, trained for 90 epochs. For FGR, the filtering ratio $\rho$ was set to 0.1. Each method was trained on three out of four domains (Art, Clipart, Real World, Product), with the remaining domain serving as an out-of-distribution test domain. We report the average across these four experimental groups. Results demonstrate that FGR provides consistent calibration gains even for data exhibiting semantic distribution shifts.
>
> | ID              | ACC    | ECE        | TS-ECE     | ACE          | TS-ACE       |
> | --------------- | ------ | ---------- | ---------- | ------------ | ------------ |
> | Cross Entropy   | 0.6322 | 0.1863     | 0.0312     | 0.005273     | 0.004677     |
> | Focal Loss      | 0.6344 | 0.0695     | 0.0334     | 0.004409     | 0.004327     |
> | Dual Focal Loss | 0.6306 | 0.0893     | 0.0321     | 0.004571     | 0.004384     |
> | BSCE-GRA        | 0.6181 | 0.0656     | 0.0376     | 0.004478     | 0.004350     |
> | FGR             | 0.6319 | **0.0632** | **0.0296** | **0.004383** | **0.004313** |
>
> | OOD             | ACC    | ECE        | TS-ECE     | ACE          | TS-ACE       |
> | --------------- | ------ | ---------- | ---------- | ------------ | ------------ |
> | Cross Entropy   | 0.3420 | 0.3645     | 0.1511     | 0.012378     | 0.008818     |
> | Focal Loss      | 0.3331 | 0.2292     | 0.1703     | 0.009954     | 0.009033     |
> | Dual Focal Loss | 0.3417 | 0.2291     | 0.1451     | 0.009746     | **0.008495** |
> | BSCE-GRA        | 0.3255 | 0.2109     | 0.1529     | 0.009911     | 0.008905     |
> | FGR             | 0.3403 | **0.2041** | **0.1393** | **0.009706** | 0.008667     |
>
> **Questions: Unclear between calibration and solutions. (W1, W2)**
>
> We hope our responses to **W1** and **W2** will resolve your questions about the relationship between calibration and solutions.

---

> > ### Comment · Reviewer_AK1A · 2025-11-24
> >
> > Thank you for your reply. I have a few more questions. I understand the motivation for applying DCT. However, this motivation seems applicable to any task, not just calibration. For example, based on your motivation, I think it would be reasonable to apply it to image classification tasks as well. So, could you explain the novelty or motivation of applying DCT to calibration tasks? Furthermore, I believe gradient correction can also be applied to any task, not just calibration.
> >
> > Moreover, ``When distributions shift, these shortcut features exhibit the most statistical variability, leading models to generate distorted high-confidence predictions based on unstable signals." Are there any experiments to support this view? I also don't know what the meaning of  ``the most statistical variability"

---

### Official Review · Reviewer_mUQE · 2025-10-31

**Soundness:** 3
**Presentation:** 3
**Contribution:** 2
**Rating:** 4
**Confidence:** 4

**Summary:**

The paper combines a frequency based filtering mechanism and a gradient rectification technique to improve the out-domain and in-domain calibration performance. Some randomly sampled images are low-pass filtered to suppress spurious correlations and then combined with original unfiltered images to compute the standard classification loss and corresponding (main) gradient. The mini-batch of original images are used to compute the soft-ECE loss and its (calib) gradient. If the main gradient conflicts with the calib gradient, then the former is projected to a hyperplane that is orthogonal to the calib gradient. This gradient projection acts as hard ID calibration constraint to sustain the ID calibration performance, which can get compromised if only filtered images are used to obtain supervisory signal for training the network. Results on different in-domain and out-domain scenarios claim to improve calibration performance.

**Strengths:**

- The challenge of improving out-domain calibration performance while also retaining the ID calibration performance is quite relevant due to its practical significance.

- Putting ID calibration as a hard constraint via gradient projection to sustain ID calibration performance is interesting and requires no weighting parameters.

- The related work provides a good coverage of recent and relevant methods in model calibration.

- The experimental comparison is shown with different baselines and in different ID and OOD scenarios and the results claim to show that the proposed methods reduces miscalibration.

**Weaknesses:**

- To compute the main gradient, why both low-pass filtered and original images are put as hybrid?

- Is it possible to use some other loss than Soft-ECE and expect similar or even better OOD and ID calibration performance?

- The weighted sum in Table 3 perform very closely to the FGR idea. Is there any explanation to that?

- It is not obvious how filtering the rectification improves OOD calibration performance?

- The ID calibration performance is not better than other methods in many cases (Table 2). Furthermore, without temperature scaling, it is not the best in any case. Given that the core hypothesis of the paper is sustaining/improving ID calibration performance and the gradient rectification is primarily proposed for this, how the Table 2 results justify this critical point?

- How the Fig. 5 visualizations are relevant to ID and OOD calibration performance?

- There are no real stats. or empirical analyses which validate the notable occurrence of gradient conflicts and connects it with the ID and OOD calibration improvements.

**Questions:**

- The core idea explicitly encourages learning of domain-invariant features, it should also help in improving out-domain accuracy notably, but this doesn't seem to be the case. Related to this, can the ID calibration hard constraint be counterproductive to improving out-of-domain accuracy in any case?

- It is not clear why the results are shown with temperature scaling in Table 1? given that the proposed method is primarily a train-time calibration contribution.

- How the method's performance is sensitive to different mini-batch size variations? Given that, the core idea is based on gradient rectification, it would be interesting to see the trend.

---

> ### Author Response · Authors · 2025-11-24
> **Author response to Reviewer mUQE - Part 1**
>
> We truly thank you for your insightful feedback and acknowledge the significance of our goal to maintain calibration performance out-of-domain, and the wide-ranging experimental content. Below, we have made our best effort to address your questions regarding the methodological principles and deeper experimental analysis.
>
> **W1: To compute the main gradient, why both low-pass filtered and original images are put as hybrid?**
>
> Before examining the model's calibration performance, we must first ensure its good classification accuracy. Therefore, the objective of $L_{main}$ is to learn a general classifier on the hybrid set, compelling the model to leverage low-frequency information more heavily for predictions while maintaining robustness to both raw and filtered inputs. This is one of the key reasons our method achieves superior calibration performance in both ID and distribution shift scenarios.
>
> **W2: Is it possible to use some other loss than Soft-ECE?**
>
> Yes, our FGR framework is **decoupled** from the calibration loss. The FGR framework fully supports replacement with other differentiable calibration losses, such as S-AvUC, MMCE, and Brier-Score. We will explicitly state this in the main text of the paper and supplement the results with experiments using the replacement losses.
>
> We present results from experiments conducted on Cifar10/100 using ResNet50, where only $L_{calib}$ was replaced and training was performed via Two-stage Fine-tuning. The results demonstrate that as a framework, FGR is not limited to using Soft-ECE. Employing other calibration optimization losses can also significantly improve calibration during distribution shifts while maintaining in-distribution performance.
>
> | Cifar10 / ResNet50 | ACC   | ECE         | TS-ECE      | ACE         | TS-ACE      |
> | ------------------ | ----- | ----------- | ----------- | ----------- | ----------- |
> | Cross-Entropy      | 95.27 | 0.04172     | 0.01143     | 0.00696     | 0.00580     |
> | FGR (Soft-ECE)     | 95.11 | **0.00863** | 0.00660     | 0.00534     | **0.00485** |
> | FGR (Soft-AvUC)    | 95.06 | 0.06068     | 0.01212     | 0.01299     | 0.00513     |
> | FGR (MMCE)         | 95.16 | 0.01062     | 0.00789     | **0.00523** | 0.00583     |
> | FGR (Brier-Score)  | 95.14 | 0.02765     | **0.00524** | 0.00794     | **0.00485** |
>
> | Cifar10-C / ResNet50 | ACC   | ECE         | TS-ECE      | ACE         | TS-ACE      |
> | -------------------- | ----- | ----------- | ----------- | ----------- | ----------- |
> | Cross-Entropy        | 74.59 | 0.22599     | 0.15197     | 0.04228     | 0.03448     |
> | FGR (Soft-ECE)       | 75.18 | 0.06881     | 0.07574     | **0.02717** | 0.02735     |
> | FGR (Soft-AvUC)      | 75.13 | 0.06471     | 0.07216     | 0.02971     | **0.02684** |
> | FGR (MMCE)           | 75.27 | 0.08124     | **0.07112** | 0.02755     | 0.02727     |
> | FGR (Brier-Score)    | 75.13 | **0.05888** | 0.07934     | 0.02728     | 0.02712     |
>
> | Cifar100 / ResNet50 | ACC   | ECE         | TS-ECE      | ACE         | TS-ACE      |
> | ------------------- | ----- | ----------- | ----------- | ----------- | ----------- |
> | Cross-Entropy       | 78.04 | 0.17109     | 0.02401     | 0.00190     | 0.00184     |
> | FGR (Soft-ECE)      | 77.79 | **0.02525** | **0.02273** | **0.00182** | 0.00170     |
> | FGR (Soft-AvUC)     | 77.84 | 0.03277     | 0.02493     | 0.00198     | **0.00168** |
> | FGR (MMCE)          | 77.87 | 0.02622     | 0.02763     | 0.00183     | 0.00189     |
> | FGR  (Brier-Score)  | 77.90 | 0.03018     | 0.02632     | 0.00194     | 0.00176     |
>
> | Cifar100-C /ResNet50 | ACC   | ECE         | TS-ECE      | ACE         | TS-ACE      |
> | -------------------- | ----- | ----------- | ----------- | ----------- | ----------- |
> | Cross-Entropy        | 51.03 | 0.38286     | 0.14109     | 0.00679     | 0.00481     |
> | FGR (Soft-ECE)       | 51.33 | 0.09934     | 0.10748     | **0.00448** | **0.00448** |
> | FGR (Soft-AvUC)      | 51.26 | **0.09538** | 0.11685     | 0.00457     | 0.00458     |
> | FGR (MMCE)           | 51.3  | 0.1096      | **0.10539** | 0.00456     | 0.00456     |
> | FGR  (Brier-Score)   | 51.15 | 0.09829     | 0.11067     | 0.00460     | 0.00460     |
>
> **W3: The weighted sum in Table 3 perform very closely to the FGR idea. Is there any explanation to that?**
>
> Thank you for your thorough review. The ‘Weighted Sum’ baseline implemented in Table 3 was trained with constant weights assigned to the main classification loss and ID calibration loss. This approach essentially seeks a compromise between the two objectives, allowing gains in classification performance at the cost of ID calibration. This contrasts with FGR, which treats ID calibration as a hard constraint only when conflicts arise.
>
> Overall, FGR's advantages include: 1. Adaptive dynamic conflict detection and projection, rather than fixed weights. 2. ‘Weighted Sum’ performance relies on careful parameter tuning, while FGR is parameter-free.

---

> ### Author Response · Authors · 2025-11-24
> **Author response to Reviewer mUQE - Part 2**
>
> **W4: How filtering the rectification improves OOD calibration performance?**
>
> Please allow us to clarify the relationship between **filtering**, **gradient rectification**, and improved OOD calibration.
>
> Previous research indicates that neural networks tend to seek simple solutions for classification problems, often exploiting notable frequency features as spurious shortcuts [1]. When distributions shift, these shortcut features exhibit the most statistical variability, leading models to generate distorted high-confidence predictions based on unstable signals. Furthermore, many common distribution shifts (e.g., corruption in Cifar-C, Fig. 1) primarily alter high-frequency components of images [2].
>
> Therefore, by applying low-pass processing (DCT filtering) to a subset of training samples, we suppressed high-frequency shortcuts and encouraged the model to rely more on stable low-frequency cues. This approach achieved significant improvements in shift scenario calibration without accessing the target domain. However, this filtering may also lead to overly conservative in-distribution confidence due to partial information loss. To maintain in-distribution calibration performance, we further resolved this conflict through our proposed theoretically guaranteed gradient rectification.
>
> **W5: About In-Distribution calibration performance.**
>
> We must restate our claim. The core contribution of our paper is not “achieving optimal in-distribution calibration,” but rather “enhancing distribution shift calibration while resolving its trade-off with in-distribution calibration”. Our approach achieves the best distribution shift calibration performance in the majority of experiments (including synthetic and real-world scenarios, Table 1) while maintaining ID calibration comparable to state-of-the-art methods (Table 2).
>
> Furthermore, as a widely used post-processing calibration technique, we demonstrate strong compatibility with temperature scaling to demonstrate FGR's practicality.
>
> **W6: How the Fig. 5 visualizations are relevant to ID and OOD calibration performance?**
>
> Figure 5 provides visual evidence supporting our core claim (W4 above). It demonstrates that FGR indeed exhibits improved learning to ignore spurious features compared to baseline BSCE-GRA, which ability explains why our framework achieves better calibration on distribution-shifted data.
>
> **W7: On statistics validating gradient conflicts and their link to calibration.**
>
> This is an excellent suggestion. We have conducted analytical experiments to statistically track gradient conflict frequency per epoch during training. We log results achieved on CIFAR-10/100 using a two-stage fine-tuning strategy with learning rate decay at epochs 390 and 420.
>
> The anonymous links for the images are as follows:
> * https://i.postimg.cc/5jcVFWft/conflict_ece_acc_over_training_Cifar10_450.png
> * https://i.postimg.cc/JtwLBCR4/conflict_ece_acc_over_training_Cifar100_450.png
>
> The conflict rate-ECE line chart clearly demonstrates: within the FGR framework, both gradient conflict rate and ECE rapidly decrease during the early training phase when the learning rate is high, perfectly validating our framework's effectiveness. However, as the learning rate decreases during the finer-tuning phase, the model tends to make the prediction distribution extremely sharp, leading to increased conflicts with the calibration loss. Our proposed gradient correction effectively prevents the model from becoming overconfident during this stage, keeping ECE consistently low.
>
> References:
>
> [1] Wang et al., What do neural networks learn in image classification?  A frequency shortcut perspective. In Proceedings of the IEEE/CVF International Conference on Computer Vision, pp. 1433–1442, 2023.
>
> [2] Zhang et al., Adversarial Defense by Suppressing High-frequency Components. arXiv preprint arXiv:1908.06566, 2019.

---

> ### Author Response · Authors · 2025-11-24
> **Author response to Reviewer mUQE - Part 3**
>
> **Q1: Does learning domain-invariant features help improve accuracy out of domain?**
>
> Our primary objective is to improve model calibration performance by encouraging the model to reduce reliance on high-frequency shortcut features through low-pass filtering. This prevents the model from generating distorted high-confidence predictions based on these unstable signals. Consequently, in the Camelyon17 dataset, the bias primarily stems from varying staining styles across hospitals, concentrated at the high-frequency/texture level. FGR enhances calibration capability while filtering out these interferences, focusing more on invariant tissue structures and achieving a significant accuracy improvement (86.83% → 89.19%). However, other datasets also contain uncommon low-frequency disruptions [3], resulting in limited overall accuracy gains.
>
> Regarding “ID calibration hard constraint” and “out-of-domain accuracy,” our ablation experiments in Table 3 of the main text demonstrate that the “ID calibration hard constraint” does not significantly impact out-of-domain accuracy. We merely projected the main loss when conflicts were detected, without substantially altering its descent direction, thus exerting minimal influence on accuracy.
>
> **Q2: Why use temperature scaling to evaluate a training-time calibration method?**
>
> We included temperature scaling (TS) results for three reasons:
>
> 1. As an easy to implement post-processing method, TS demonstrated great practicality in in-distribution scenarios.
> 2. Previous works [4][5] have almost always reported performance using TS in combination with baselines, so we wanted to present a more comprehensive comparison.
> 3. We aim to demonstrate that our training framework FGR and the post-processing method are complementary. FGR improves the model itself, while TS can further refine the results built upon it.
>
> **Q3: How the method's performance is sensitive to different mini-batch size variations?**
>
> Thanks, this is an interesting technical question. We have conducted sensitivity analysis experiments related to batch size (bs) on Cifar10/100 and their corresponding corrupted datasets. Our baseline batch size is set to 128. Our baseline batch size is set to 128. Since directly reducing the batch size leads to a significant drop in accuracy, we employ 4×/2× gradient accumulation for bs=32/64 to maintain equivalent optimization strides. Experimental results are presented in the table below.
>
> We observe that: Since FGR's gradient rectification is based on conflict detection across the entire batch, smaller batch sizes (e.g., bs=32, 64) capture more fine-grained conflicts between the main loss and ID calibration loss. This yields the most significant improvement in OOD clibration performance while maintaining in-distribution performance. Larger batch sizes (e.g., ba=512) may hide certain local conflicts, leading to a clear performance drop. Although smaller batches perform better, training time also increases (32 vs. 128 → 4.83h vs. 1.7h). To achieve the optimal balance, we still recommend using a medium-sized batch (i.e., bs=128). We will incorporate these relevant discussions into the main text of the paper.
>
> * **Cifar10:**
>
> | bs   | ACC| ECE| TS-ECE| ACE| TS-ACE |
> |-|-|-|-|-|-|
> | 32   | 94.26 | 0.01367 | 0.01135     | 0.00388 | 0.00306|
> | 64   | 94.23 | **0.00803** | 0.00836     | 0.00301 | 0.00306  |
> | 128  | 94.09 | 0.00819  | 0.00722     | 0.00328     | 0.00326     |
> | 256  | 94.30 | 0.01347 | **0.00719** | **0.00231** | **0.00217** |
> | 512  | 94.10 | 0.02253 | 0.01283 | 0.00268     | 0.00237     |
>
> * **Cifar10-C:**
>
> | bs|ACC| ECE| TS-ECE| ACE | TS-ACE|
> |-| - | - |- |-|-|
> | 32   | 77.01 | **0.07763** | 0.10432| **0.02795** | **0.02911** |
> | 64   | 74.21 | 0.09988| **0.09802** | 0.03266| 0.03258|
> | 128  | 73.57 | 0.12203| 0.11628| 0.03556 | 0.03525|
> | 256  | 73.14 | 0.13267| 0.11811 | 0.03602| 0.03506|
> | 512  | 72.52 | 0.14811| 0.12334| 0.03737| 0.03556|
>
> * **Cifar100:**
>
> | bs   | ACC   | ECE | TS-ECE | ACE| TS-ACE|
> |-|-|-|-|-|-|
> |32| 78.14 | **0.01417** | **0.01470** | 0.00130| 0.00138 |
> |64| 77.92 | 0.01735| 0.02126| 0.00134| 0.00143|
> |128| 78.35 | 0.01970| 0.01790| 0.00185| 0.00181|
> |256| 77.75 | 0.01759| 0.01534| **0.00093** | **0.00115** |
> |512| 76.24 | 0.03181| 0.02104| 0.00105 | 0.00123|
>
> * **Cifar100-C:**
>
> |bs|ACC|ECE|TS-ECE| ACE|TS-ACE|
> |-|-|-|-|-|-|
> |32| 52.58 | 0.10546| 0.09711| **0.00477** | **0.00478** |
> |64| 51.87 | 0.10457| **0.09660** | 0.00484| 0.00697|
> |128| 51.02 | **0.10145** | 0.10562| 0.00495| 0.00495|
> |256| 51.37|0.11751| 0.10495| 0.00484| 0.00483|
> |512| 48.63|0.14406| 0.10356| 0.00518| 0.00511|
>
> References:
>
> [3] Vaish et al., Fourier-basis functions to bridge augmentation gap: Rethinking frequency augmentation in image classification, *ICCV*, 2024.
>
> [4] Karandikar et al., Soft calibration objectives for neural networks. *NeurIPS*, 2021.
>
> [5] Neo et al., Maxent loss: Constrained maximum entropy for calibration under out-of-distribution shift. *AAAI*, 2024.

---

> > ### Comment · Reviewer_mUQE · 2025-11-26
> > **Rebuttal acknowledgement and further questions**
> >
> > I thank authors for posting responses to all of my comments. Below, I will mention those responses for which I am not satisfied and seek further clarifications:
> >
> > W5: ID calibration performance: I understand that the main claim of the submission is not achieving optimal ID calibration performance, however, the issue is that the performance of the method, especially without TS, is mostly poor from best performing methods (Table 2). Now, how does this satisfy the ID vs OOD calibration tradeoff, and really why one would prefer this method when it is improving calibration performance in OOD scenarios at the cost of mostly poor ID calibration performance?
> >
> > W6: Fig. 5 visualizations: I don't think the current explanation is clear enough. The main goal of the approach is not reducing reliance of spurious features, instead relying on them for gradient rectification whenever there is a conflict.
> >
> > W3: Weighted sum performance: I wonder if those fixed weights were tuned using some validation set or so?
> >
> > Q3: Batchsize vs performance: This a bit strange that the performance drops with increasing batch size. I don't get much how they hide gradient conflicts as such?

---

### Official Review · Reviewer_EPQx · 2025-11-01

**Soundness:** 2
**Presentation:** 2
**Contribution:** 2
**Rating:** 4
**Confidence:** 3

**Summary:**

The paper proposes Frequency-aware Gradient Rectification (FGR), a training-time framework to enhance calibration of deep neural networks under distribution shift, without requiring any target domain access. The approach combines low-pass discrete cosine transform (DCT) filtering of part of the training data, intended to encourage reliance on domain-invariant, low-frequency features, with a geometric gradient rectification step that projects parameter updates to prevent an increase in in-distribution (ID) calibration error. The authors support their method with theoretical analysis, comprehensive empirical studies on CIFAR-C variants, Tiny-ImageNet-C, and WILDS benchmarks, and ablation/visualization experiments.

**Strengths:**

1. The use of DCT-based block-wise low-pass filtering is a creative tool for encouraging robustness to shift, going beyond standard pixel-level or data augmentation strategies.
2. The proposed projection-based gradient rectification is well-motivated and avoids hyperparameter tuning.

**Weaknesses:**

1. The proposed method improves upon the compared method for metrics except accuracy.
2. Poor performance on most metrics for the TinyImageNet dataset, including in the results in the appendix. This reflects poorly on the efficacy of the proposed approach.

**Questions:**

1. Can the authors justify why the proposed approach performs not as well on the TinyImageNet dataset?
2. Can the authors discuss how the accuracy of the proposed method can be improved through any trade-off.

---

> ### Author Response · Authors · 2025-11-24
> **Author response to Reviewer EPQx**
>
> Thank you for your thorough review. We are pleased that you recognize the innovative nature of the DCT low-pass filtering strategy in enhancing offset robustness. Below is our point-by-point response to your questions:
>
> **Q1/W2: Why the performance on TinyImageNet is not as well?**
>
> Tiny-Imagenet, as a dataset comprising 200 categories but with lower image resolution (only 64×64), is more fine-grained compared to CIFAR-10/100. To test the generalizability of our method and ensure reproducibility across datasets, we applied the same filter strength $\lambda$ to all datasets (no per-dataset tuning). However, this may have resulted in excessive removal of useful frequency information within Tiny-Imagenet, leading to a less pronounced improvement in FGR on this dataset. Nevertheless, its performance remains broadly comparable to other baselines.
>
> We attempted to adjust the filtering strength $\lambda$ on the TinyImageNet dataset to validate this assumption. During training, FGR sets the filtering strength $\lambda$ of filtered samples to be taken from a range of [$\lambda_{min}$, 25] (where smaller $\lambda$ means more high-frequency information is filtered out). In previous experiments, we set a default $\lambda_{min}=15$ for all datasets. Here, we modified $\lambda_{min}\in \\{15,18,25\\}$ to conduct experiments with training from scratch, keeping all other settings unchanged. The experimental results are shown in the table below. These results indicate that by appropriately adjusting the filtering strength, the performance of FGR still has significant improvement potential. It can achieve optimal out-of-distribution calibration performance on Tiny-Imagenet-C while maintaining good in-distribution calibration.
>
> | | ID-ACC$\uparrow$ | ID-ECE$\downarrow$ | ID-ACE$\downarrow$ | OOD-ACC$\uparrow$ | OOD-ECE$\downarrow$ | OOD-ACE$\downarrow$ |
> | - | -- | -|-|-|-|- |
> | CE| **65.39** | 0.13598  | 0.00081   | **24.29**  | 0.35522 | 0.00491   |
> | FLSD-53  | 60.26  | **0.01957**        | 0.00081    | 21.90    | 0.17136             | 0.00425             |
> | DFL | 64.64 | 0.02831            | 0.00064            | 23.67             | 0.20314             | 0.00431             |
> | BSCE-GRA | 63.80            | 0.02027            | 0.00066            | 22.95             | 0.20423             | 0.00450             |
> | FGR($\lambda_{min}=15$, High) | 64.29            | 0.02760            | 0.00067            | 22.57             | 0.15050             | 0.00436             |
> | FGR($\lambda_{min}=18$, Mid)  | 64.08            | 0.02261            | **0.00062**        | 23.38             | **0.14095**         | **0.00424**         |
> | FGR($\lambda_{min}=25$, Low)  | 64.03            | 0.02625            | 0.00065            | 23.06             | 0.14887             | 0.00428             |
>
> **Q2/W1: Can the authors discuss how the accuracy of the proposed method can be improved through any trade-off.**
>
> Within the FGR framework, we can bias in-distribution accuracy by adjusting the filtering ratio $\rho$ or filtering strength ($\lambda$) (as demonstrated in Appendix D.1 Hyperparameter sensitivity analysis). Our method provides tools to manage this trade-off. Specifically, reducing the proportion of filtered images $\rho$ allows the model to observe more unfiltered images. Increasing $\lambda$ to weaken the filtering strength preserves more high-frequency details, bringing images closer to the original distribution and thus better preserving in-distribution accuracy. However, our extensive experiments demonstrate that under appropriate settings, FGR's impact on in-distribution accuracy is negligible.
>
> Beyond this, we must clarify that the primary objective of the proposed FGR is to optimize calibration performance both in-distribution and during distribution shifts without decreasing accuracy, similar to other calibration methods \[1][2]. It is not intended to achieve state-of-the-art accuracy on shifted data. Table 1 in Section 5 demonstrates that FGR significantly improves calibration performance while maintaining competitive out-of-distribution accuracy. Previous similar methods like Augmix [3] demonstrated outstanding performance on in-distribution data but suffered significant accuracy drops in real-world shifted scenarios due to overly strong data augmentation. On Camelyon17, our method (89.19%) also outperforms CE (86.83%) in accuracy.
>
> [1] Karandikar et al., Soft calibration objectives for neural networks. Advances in Neural Information Processing Systems, 34:29768–29779, 2021.
>
> [2] Neo et al., Maxent loss: Constrained maximum entropy for calibration under out-of-distribution shift. In Proceedings of the AAAI conference on artificial intelligence, volume 38, pp. 21463–21472, 2024.
>
> [3] Hendrycks et al., Augmix: A simple method to improve robustness and uncertainty under data shift. In International Conference on Learning Representations, 2020.

---

### Note · Authors · 2025-12-02

I have read and agree with the venue's withdrawal policy on behalf of myself and my co-authors.